# Interpretable Meta-Learning of Physical Systems

**Matthieu Blanke**
Inria Paris, DI ENS, PSL Research University
`matthieu.blanke@inria.fr`

**Marc Lelarge**
Inria Paris, DI ENS, PSL Research University
`marc.lelarge@inria.fr`

## Abstract

Machine learning methods can be a valuable aid in the scientific process, but they need to face challenging settings where data come from inhomogeneous experimental conditions. Recently, meta-learning approaches have made significant progress in multi-task learning, but they rely on black-box neural networks, resulting in high computational costs and limited interpretability. Leveraging the structure of the learning problem, we argue that multi-environment generalization can be achieved using a simpler learning model, with an affine structure with respect to the learning task. Crucially, we prove that this architecture can identify the physical parameters of the system, enabling interpretable learning. We demonstrate the competitive generalization performance and the low computational cost of our method by comparing it to state-of-the-art algorithms on physical systems, ranging from toy models to complex, non-analytical systems. The interpretability of our method is illustrated with original applications to parameter identification and to adaptive control.

## 1 Introduction

Learning physical systems is an essential application of artificial intelligence that can unlock significant technological and societal progress. Physical systems are inherently complex, making them difficult to learn Karniadakis et al. (2021). A particularly challenging and common scenario is multi-environment learning, where observations of a physical system are collected under inhomogeneous experimental conditions Caruana (1997). In such cases, the scarcity of training data necessitates the development of robust learning algorithms that can efficiently handle environmental changes and make use of all available data.

This multi-environment learning problem falls within the framework of multi-task learning, which has been widely studied in the field of statistics since the 1990s (Caruana, 1997). The aim is to exploit task diversity to learn a shared representation of the data and thus improve generalization. With the rise of deep learning, several meta-learning approaches have attempted in recent years to incorporate multi-task generalization into gradient-based training of deep neural networks. In the seminal paper by Finn et al. (2017), and several variants that followed (Zintgraf et al., 2019; Raghu et al., 2020), this is done by integrating an inner gradient loop in the training process. Alternatively, Bertinetto et al. (2019) proposed adapting the weights using a closed-form solver. As far as physical systems are concerned, the majority of the proposed methods have focused on specific architectures oriented towards trajectory prediction (Wang et al., 2022a; Kirchmeyer et al., 2022).

When learning a physical system from data, a critical yet often overlooked challenge is model interpretability (Lipton, 2018; Grojean et al., 2022). Interpreting the learned parameters in terms of the system's physical quantities is crucial to making the model more explainable, allowing for scientific discovery and downstream model-based applications such as control. The above approaches benefit from the expressiveness of deep learning, but are costly in terms of computational time, both for learning and for inference. Furthermore, the complexity and the black-box nature of neural networks hinder the interpretability of the learned parameters, even when the physical system is linearly parametrized.

Recently, Wang et al. (2021) showed theoretically that the learning capabilities of gradient-based meta-learning algorithms could be matched by the simpler architecture of multi-task representation

learning with hard parameter sharing, where the heads of a neural network are trained to adapt to multiple tasks (Caruana, 1997; Ruder, 2017). They also demonstrated empirically that this architecture is competitive against state-of-the-art gradient-based meta-learning algorithms for few-shot image classification. We propose to use multi-task representation learning for physical systems, and show how it can bridge the gap between the power of neural networks and the interpretability of the model, with minimal computational costs.

**Contributions** In this work, we study the problem of multi-environment learning of physical systems. We model the variability of physical systems with a multi-task representation learning architecture that is affine in task-specific parameters. By exploiting the structure of the learning problem, we show how this architecture lends itself to multi-environment generalization, with considerably lower cost than complex meta-learning methods. Additionally, we show that it enables identification of physical parameters for linearly parametrized systems, and local identification for arbitrary systems. Our method's generalization abilities and computational speed are experimentally validated on various physical systems and compared with the state of the art. The interpretability of our model is illustrated by applications to parameter identification and to adaptive control.

## 2 LEARNING FROM MULTIPLE PHYSICAL ENVIRONMENTS

In this section, we present the problem of multi-task learning as it occurs in the physical sciences and we summarize how it can be tackled with deep learning in a meta-learning framework.

### 2.1 THE VARIABILITY OF PHYSICAL SYSTEMS

In general, a physical system is not fixed from one interaction to the next, as experimental conditions vary, whether in a controlled or uncontrolled way. From a learning perspective, we assume a meta-dataset $D := \cup_{t=1}^T D_t$ composed of $T$ datasets, each dataset gathering observations of the physical system under specific experimental conditions. The goal is to learn a predictor from $D$ that is robust to task changes, in the sense that when presented a new task, it can learn the underlying function from a few samples (Hospedales et al., 2021). Note that in practice the number of tasks $T$ is typically very limited, owing to the high cost of running physical experiments.

For simplicity, we assume a classical supervised regression setting where $D_t := \{x_t^{(i)}, y_t^{(i)}\}_{1 \leq i \leq N_t}$ and the goal is to learn a $x \mapsto y$ predictor, although the approaches presented generalize to other settings such as trajectory prediction of dynamical systems. We discuss two physical examples illustrating the need for multi-task learning algorithms, with different degrees of complexity.

**Example 1** (Actuated pendulum). We begin with the pendulum, one of physics' most famous toy systems. Denoting its inertia and its mass by $I$ and $m$ and the applied torque by $u$, the angle $q$ obeys

$$I\ddot{q} + mg\sin q = u. \tag{2.1}$$

For example, we may want to learn the action $y = u$ as a function of the coordinates $x = (q, \dot{q}, \ddot{q})$. In a data-driven framework, the trajectories collected may show variations in the pendulum parameters: the same equation (2.1) holds true, albeit with different parameters $m$ and $I$.

A more complex, non-analytical example is that of learning the solution to a partial differential equation, which is rarely known in closed form and varies strongly according to the boundary conditions.

**Example 2** (Electrostatic potential). The electrostatic potential $y$ in a space $\Omega$ devoid of charges solves Laplace's equation, with boundary conditions

$$\Delta y = 0 \quad \text{on } \Omega, \qquad y(x) = b(x) \quad \text{on } \partial\Omega. \tag{2.2}$$

A robust data-driven solver should be able to generalize to (at least small) changes of $\partial\Omega$ and $b$.

### 2.2 OVERVIEW OF MULTI-ENVIRONMENT DEEP LEARNING

Multi-task statistical learning has a long history, and several approaches to this problem have been proposed in the statistics community (Caruana, 1997). We will focus on the meta-learning paradigm (Hospedales et al., 2021), which has recently gained considerable importance and whose application to neural nets looks promising given the complexity of physical systems. We describe the generic structure of meta-learning algorithms for multi-task generalization.

**Learning model**  Given the learning capabilities of neural networks, incorporating multi-task generalization into their gradient descent training algorithms is a major challenge. Since the seminal paper by Finn et al. (2017), several algorithms have been proposed for this purpose, with the common idea of finding a map adapting the weights of the neural network according to task data. A convenient point of view is to introduce a two-fold parametrization of a meta-model $F(x; \theta, w)$, with a task-agnostic parameter vector $\theta \in \mathbb{R}^p$ and task-specific weights $w$ (also called learning contexts). For each task $t$, the task-specific weight is computed based on some trainable meta-parameters $\pi$ and the task data currently being processed as $w_t := A(\pi, D_t)$, according to an adaptation rule $A$ that is differentiable with respect to $\pi$. The meta-parameters are trained to minimize the meta-loss function aggregated over the tasks, as we will see below.

We provide examples of recent architectures in Table 1. In MAML (Finn et al., 2017), the meta-parameter $\pi$ is simply $\theta$ and the adaptation rule is computed as a gradient step in the direction of the task-specific loss improvement, in an inner gradient loop. In CoDA (Kirchmeyer et al., 2022), the meta-parameter $\pi$ has a dimension growing with the number of tasks $t$ and the adaptation rule is computed directly from the meta-parameters, with task-specific low-dimensional context vectors $\xi_t \in \mathbb{R}^{d_\xi}$ and a linear hypernetwork $\Theta \in \mathbb{R}^{p \times d_\xi}$. Variants of MAML, CAVIA (Zintgraf et al., 2019) and ANIL (Raghu et al., 2020), fit into this scheme as well and correspond to the restriction of the adaptation inner gradient loop to a predetermined set of the network's weights. This framework also encompasses the CAMEL algorithm, which we introduce in Section 3.

**Meta-training**  The training process is summarized in Algorithm 1. For each task $t$, the meta-learner computes a task-specific version of the model from the task dataset $D_t$, defining $f_t(x; \pi) := F(x; \theta, A(\pi, D_t))$. The error on the dataset $D_t$ is measured by the task-specific loss

$$\ell(D_t; \theta, w) = \sum_{x, y \in D_t} \frac{1}{2} \big( F(x; \theta, w) - y \big)^2. \tag{2.3}$$

Parameters $\pi$ are trained by gradient descent in order to minimize the regularized meta-loss defined as the aggregation of $L_t$ and a regularization term $R(\pi)$:

$$L(\pi) := \sum_{t=1}^{T} \ell \big( D_t; \theta, w_t(\pi) \big) + R(\pi). \tag{2.4}$$

---

**Algorithm 1** Gradient-based meta-training

**input** meta-model $F(x; \theta, w)$, adaptation rule $A$, initial meta-parameters $\pi$, learning rate $\eta$, task datasets $D_1, \dots D_T$
**output** learned meta-parameters $\bar{\pi}$
**while** not converged **do**
    **for** tasks $1 \leq t \leq T$ **do**
        compute $\theta$ from $\pi$
        adapt $w_t(\pi) := A(\pi, D_t)$
        compute $\ell \big( D_t; \theta, w_t(\pi) \big)$
    **end for**
    compute $L(\pi)$, as in (2.4)
    update $\pi \leftarrow \pi - \eta \nabla L(\pi)$
**end while**

Table 1: Structure of various meta-learning models. Here $h(x; \theta) \in \mathbb{R}$ and $v(x; \theta) \in \mathbb{R}^r$ denote arbitrary parametric models, such as neural networks; "order" stands for differentiation order.

|  | MAML | CoDA | CAMEL |
|---|---|---|---|
| $\pi$ | $\theta$ | $\theta, \Theta, \{\xi_t\}$ | $\theta, \{\omega_t\}$ |
| $\dim(\pi)$ | $p$ | $p + p \times d_\xi + d_\xi \times T$ | $p + r \times T$ |
| $\dim(w)$ | | $p$ | $r$ |
| $A(\pi, D_t)$ | $-\alpha \nabla_\theta L_t$ | $\Theta \xi_t$ | $\omega_t$ |
| $F(x; \theta, w)$ | | $h(x; \theta + w)$ | $w^\top v(x; \theta)$ |
| training order | 2 | 1 | 1 |
| adaptation order | 1 | 1 | 0 |

**Test-time adaptation**  At test time, the trained meta-model is presented with a dataset $D_{T+1}$ consisting of few samples (or shots) from a new task. Using this adaptation data and the learned meta-parameters $\bar{\pi}$, the task-agnostic component $\bar{\theta}$ of the meta-model is frozen, and the task-specific component is tuned (possibly in a constrained set) by minimizing the prediction error on the adaptation dataset:

$$w_{T+1} \in \operatorname*{argmin}_{w} \ell \big( D_{T+1}; \bar{\theta}, w \big). \tag{2.5}$$

In all the above approaches, this minimization is performed by gradient descent. The resulting adapted predictor is defined as $F(x; \bar{\theta}, w_{T+1})$, and is evaluated by its performance on a separated test set from task $T + 1$, averaged over the task distribution.

## 3  CONTEXT-AFFINE MULTI-ENVIRONMENT LEARNING

Physical systems often have a particular structure in the form of mathematical models and equations. The general idea behind model-based machine learning is to exploit the available structure to increase learning performance and minimize computational costs (Karniadakis et al., 2021). With this in mind, we adopt in this section a simpler architecture than those shown above, and show how it lends itself particularly well to learning physical systems.

**Problem structure**  We note that many equations in physics exhibit an affine task dependence, since the varying physical parameters often are linear coefficients (as we see in Example 1, and we shall further explain in Section 4). By incorporating this same structure and hence mimicking physical equations, the model should be well-suited for learning them and for interpreting the physical parameters. Following these intuitions, we propose to learn multi-environment physical systems with affine task-specific context parameters.

**Definition 1** (Context-affine multi-task learning). The prediction is modeled as an affine function of low-dimensional task-specific weights $w \in \mathbb{R}^r$ with a task-agnostic feature map $v(x; \theta) \in \mathbb{R}^r$ and a task-agnostic bias $c(x; \theta) \in \mathbb{R}$:

$$F(x; \theta, w) = c(x; \theta) + w^\top v(x; \theta). \tag{3.1}$$

The dimension $r$ of the task weight must be chosen carefully. It must be larger than the estimated number of physical parameters varying from task to task but smaller than the number of training tasks, so as to observe the function $v$ projected over a sufficient number of directions. During training, the task-specific weights are directly trained as meta-parameters along with the shared parameter vector: $\pi = (\theta, \omega_1 \ldots, \omega_T)$ and $w_t = A(\pi, D_t) = \omega_t$. The meta-parameters are jointly trained by gradient descent as in Algorithm 1. At test time, the minimization problem of adaptation (2.5) reduces to ordinary least squares.

The architecture introduced in Definition 1 is equivalent to multi-task representation learning with hard parameter sharing Ruder (2017) and is proposed as a meta-learning algorithm in (Wang et al., 2021) We will refer to it in our physical system framework as Context-Affine Multi-Environment Learning (CAMEL). In this work, we show that CAMEL is particularly relevant for learning physical systems. Table 1 compares CAMEL with the meta-learning algorithms described above.

**Computational benefits**  As the task weights $(\omega_t)_{t=1}^T$ are kept in memory during training instead of being computed in an inner loop, CAMEL can be trained at minimal computational cost. In particular, it does not need to compute Hessian-vector products as in MAML, or to propagate gradients through matrix inversions as in (Bertinetto et al., 2019). Adaptation at test time is also computationally inexpensive since ordinary least squares guarantees a unique solution in closed form, as long as the number of samples exceeds the dimension $r$ of the task weight. For real-time applications, the online least-squares formula (Kushner & Yin, 2003) ensures adaptation with minimal memory and compute requirements, whereas gradient-based adaptation (as in CoDA or in MAML) can be excessively slow.

**Applicability**  The meta-learning models described in Section 2.2 seek to learn multi-task data from a complex parametric model (typically a neural network), making the structural assumption that the weights vary slightly around a central value in parameter space: $f_t(x; \pi) = h(x; \theta_0 + \delta\theta_t)$, with $\|\delta\theta\| \ll \|\theta_0\|$. Extending this reasoning, the model should be close to its linear approximation:

$$h(x; \theta_0 + \delta\theta_t) \simeq h(x; \theta_0) + \delta\theta_t^\top \nabla h(x; \theta_0), \tag{3.2}$$

where we observe that the output is an affine function of the task-specific component $\delta\theta_t$. We believe that (3.2) explains the observation that MAML mainly adapts the last layer of the neural network (Raghu et al., 2020). In Definition 1, $v$ and $c$ are arbitrary parametric models, which can be as complex as a deep neural network and are trained to learn a representation that is linear in the task weights. Following (3.2), we expect CAMEL's expressivity to be of the same order as that of more complex architectures, with $c(x; \theta)$, $w_t$ and $v(x; \theta)$ playing the roles of $h(x; \theta_0)$, $\delta\theta_t$ and $\nabla h(x; \theta)$ respectively. Another key advantage of CAMEL is the interpretability of the model, which we describe next.

## 4 INTERPRETABILITY AND SYSTEM IDENTIFICATION

The observations of a physical system are often known to depend on certain well-identified physical quantities that may be of critical importance in the scientific process. When modeling the system in a data-driven approach, it is desirable for the trained model parameters to be interpretable in terms of these physical quantities (Karniadakis et al., 2021), thus ensuring controlled and explainable learning (Linardatos et al., 2021). We here focus on the identification of task-varying physical parameters, which raises the question of the identifiability of the learned task-specific weights. System identification and model identifiability are key issues when learning a system (Ljung, 1998). Although deep neural networks are becoming increasingly popular for modeling physical systems, their complex structure makes them impractical for parameter identification in general (Nelles, 2001).

**Physical context identification**   In mathematical terms, the observed output is considered as an unknown function $y(x; \varphi)$ of the input and a physical context vector $\varphi \in \mathbb{R}^n$, gathering the parameters of the system. In our multi-environment setting, each task is defined by a vector $\varphi_t$ as $y_t(x) = y(x, \varphi_t)$. At test time, a new environment corresponds to an unknown underlying physical context $\varphi_{T+1}$. While adaptation consists in minimizing the prediction error on the data as in (2.5), the interpretation goes further and seeks to identify $\varphi_{T+1}$. This means mapping the learned task-specific weights $w$ to the physical contexts $\varphi$, *i.e.* learning an estimator $\hat{\varphi} : w \mapsto \varphi$ using the training data and the trained model. Assuming that the physical parameters of the training data $\{\varphi_t\}$ are known, this can be viewed as a regression problem with $T$ samples, where $\hat{\varphi}$ is trained to predict $\varphi_t$ from weights $w_t$ learned on the training meta-dataset.

### 4.1 LINEARLY PARAMETRIZED SYSTEMS

We are primarily interested in the case where the physical parameters are known to intervene linearly in the system equation, as

$$y(x; \varphi) = \kappa(x) + \varphi^\top \nu(x), \quad \nu(x) \in \mathbb{R}^n. \tag{4.1}$$

This class of systems is of crucial importance: although simple, it covers a large number of problems of interest, as the following examples illustrate. Furthermore, it can apply locally to more general system, as we shall see later.

**Example 3** (Electric point charges). Point charges are a particular case of Example 2 with point boundary conditions, proportional to the charges $\varphi = (\varphi^{(1)}, \ldots, \varphi^{(n)})$. The resulting field can be computed using Coulomb's law and is proportional to these charges: $y(x; \varphi) = \varphi^\top \nu(x)$, with $\nu(x) \propto (1/\|x - x^{(j)}\|)_j$. Although the solution is known in closed form, this example can illustrate more complex problems where an analytical solution is out of reach (and hence $\nu$ is unknown) but the linear dependence on certain well-identified parameters is postulated or known.

**Example 4** (Inverse dynamics in robotics). One application where our model is particularly well suited in robotics is inverse dynamics: it turns out that the Euler-Lagrange formulation for the rigid body dynamics is always linear with respect to the system's dynamic parameters (Nguyen-Tuong & Peters, 2010), and hence takes the form of (4.1). A simple, yet illustrative system with this structure is the actuated pendulum (2.1), where it is clear that the equation is linear in the inertial parameters $I$ and $m$. The inverse dynamics equation can be used for trajectory tracking (Spong et al., 2020), as it predicts $u$ from a target trajectory $\{q(s)\}$. We provide more details in Appendix B.3.

### 4.2 LOCALLY LINEAR PHYSICAL CONTEXTS

In the absence of prior knowledge about the system under study, the most reasonable structural assumption for multi-task data is to postulate small variations in the system parameter: $\varphi = \varphi_0 + \delta\varphi$. The learned function can then be expanded and found to be locally linear in physical contexts:

$$y(x; \varphi) \simeq y(x; \varphi_0) + \delta\varphi^\top \nabla y(x; \varphi_0), \tag{4.2}$$

which has the form (4.1) with $\kappa(x) = y(x; \varphi_0)$ and $\nu(x) = \nabla y(x; \varphi_0)$.

**Example 5** (Identification of boundary perturbations). For a general boundary value problem such as (2.2), we may assume that the boundary conditions $\partial\Omega(\varphi), b(x, \varphi)$ vary smoothly according to parameters $\varphi$ (such as angles or displacements). If these variations are small and the problem is sufficiently regular, the resulting solution $y(x, \varphi)$ can be reasonably well approximated by (4.2).

### 4.3 PARAMETER IDENTIFICATION WITH CAMEL

We now study the problem of system identification under the assumption of parameter linearity (4.1) using the CAMEL metamodel (3.1). We study the identifiability of the model and therefore investigate the vanishing training loss limit, with $c = \kappa = 0$ for simplicity, yielding

$$\omega_t^\top v(x_t^{(i)}) = \varphi_t^\top \nu(x_t^{(i)}) \quad \text{for all} \quad 1 \leq t \leq T, \ 1 \leq i \leq N_t. \tag{4.3}$$

**Identifiability**    Posed as it is, we can easily see that the physical parameters $\varphi_t$ are not directly identifiable. Indeed, for any $P \in \mathrm{GL}_r(\mathbb{R})$, the weights $\omega$ and the feature map $v$ produce the same data as the weights $\omega' := P^\top \omega$ and the feature map $v' = P^{-1}v$, since $\omega^\top v = \omega^\top PP^{-1}v$. This problem is related to that of identification in matrix factorization (see for example Fu et al. (2018)). Now that we have recognized this symmetry of the problem, we can ask whether it characterizes the solutions found by CAMEL. The following result provides a positive answer.

**Proposition 1.** Assume that the training points are uniform across tasks: $x_t^{(i)} = x^{(i)}$, and $N_t = N$ for all $1 \leq t \leq T$ and $1 \leq i \leq N$, with $n \leq r < N, T$. Assume that both sets $\{\nu(x^{(i)})\}$ and $\{\varphi_t\}$ span $\mathbb{R}^n$. In the limit of a vanishing training loss $L(\pi) = 0$, the trained meta-parameters recover the parameters of the system up to a linear transform: there exist $P, Q \in \mathbb{R}^{n \times r}$ such that $\varphi_t = P\omega_t$ for all training task $t$ and $\nu(x^{(i)}) = Qv(x^{(i)})$ for all $1 \leq i \leq N$. Additionally, $QP^\top = I_n$.

A proof is provided in Appendix A, along with the case $\kappa \neq 0$. Proposition 1 shows that CAMEL learns a meaningful representation of the system's features instead of overfitting the examples from the training tasks. Remarkably, the relationship between the learned weights and the system parameters is linear and can be estimated using ordinary least squares:

$$\hat{\varphi}(\omega) = \hat{P}\omega, \quad \hat{P} \in \underset{P \in \mathbb{R}^{n \times r}}{\operatorname{argmin}} \frac{1}{2} \sum_{t=1}^{T} \|P\omega_t - \varphi_t\|_2^2. \tag{4.4}$$

For black-box meta-learning architectures, exhibiting the symmetries in model parameters and computing an identification map seems out of reach, as the number of available tasks $T$ can be very limited in practice (Pourzanjani et al., 2017).

## 5 EXPERIMENTING ON PHYSICAL SYSTEMS

The architecture that we have presented is expected to adapt efficiently to the prediction of new environments, and identify (locally or globally) their physical parameters, as shown in Section 4. In this section, we validate these statements experimentally on various physical systems: Sections 5.1 and 5.2 deal with systems with linear parameters (as in (4.1)), on which we evaluate the interpretability of the algorithms. We then examine a non-analytical, general system in Section 5.3. We compare the performances of CAMEL with state-of-the-art meta-learning algorithms. Our code and demonstration material are available at `https://github.com/MB-29/meta-learning`.

**Baselines**    We have implemented the MAML algorithm of Finn et al. (2017), and its ANIL variant (Raghu et al., 2020), which is computationally lighter and more suitable for learning linearly parametrized systems (according to observation (3.2)). We have also adapted the $\ell_1$-CoDA architecture of Kirchmeyer et al. (2022) for supervised learning (originally designed for time series prediction). In all our experiments, the different meta-models share the same underlying neural network architecture, with the last layer of size $r \gtrsim \dim(\varphi)$. Additional details can be found in Appendix B. The linear regressor computed for CAMEL in (4.4) is computed after training for all architectures with their trained weights $w_t$, and is available at test time for identification.

### 5.1 INTERPRETABLE LEARNING OF AN ELECTRIC POINT CHARGE SYSTEM

As a first illustration of multi-environment learning, we are interested in a data-driven approach to electrostatics, where the experimenter has no knowledge of the theoretical laws (Maxwell's equations, as in Example 2) of the system under study. The electrostatic potential is measured at various points in space, under different experimental conditions. The observations collected are then used to train a meta-learning model to predict the electrostatic field from new experiments, based on very limited data. We start with the toy system described in Example 3,

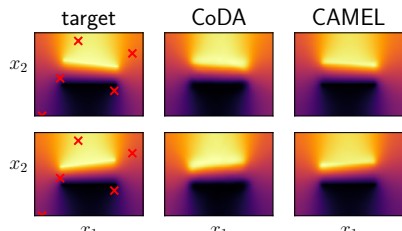

Figure 1: Few-shot adaptation on two out-of-domain environments of the point charge system in a dipolar setting (**left**) and the capacitor (**right**). The adaptation points are represented by the $\times$ symbols. The vector fields are derived from the learned potential fields using automatic differentiation.

which provides a qualitative illustration of the behavior of various learning algorithms: $n = 3$ point charges placed in the plane at fixed locations. This experiment is repeated with varying charges $\varphi \in \mathbb{R}^3$.

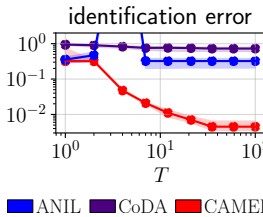

**Results**   For this system with linear physical parameters, CAMEL outperforms other baselines and can predict the electrostatic field with few shots, as shown in Figure 1 and Table 2 (5-shot adaptation). Figure 2 shows the identification error over 30 random test environments with standard deviations, as a function of the number of training tasks. Thanks to the sample complexity of linear regression, CAMEL accurately identifies system charges, achieving less than 1% relative error with 10 training tasks.

Figure 2: Average relative error for the point charge identification.

## 5.2   MULTI-TASK REINFORCEMENT LEARNING AND ONLINE SYSTEM IDENTIFICATION

Another scientific field in which our theoretical framework can be applied is multi-task reinforcement learning, in which a control policy is learned using data from multiple environments of one system (Vithayathil Varghese & Mahmoud, 2020). We saw in Example 4 that robot joints obey the inverse dynamics equation, which turns out to be linear in the robot's inertial parameters. Consequently, our architecture lends itself well to the statistical learning of this equation from multiple environment data, as well as to the identification of the dynamic parameters. We may then exploit the learned model of the dynamics to perform adaptive inverse dynamics control (see Appendix B.4) of robots with unknown parameters, and identify the parameters simultaneously.

**Systems**   We experiment with systems of increasing complexity, starting with 2D simulated systems: cartpole and acrobot. To make them more realistic, we add friction in their dynamics. The analytical equation (4) is hence inaccurate, which motivates the use of a data-driven learning method. We then experiment on the simulated 6-degree-of-freedom robot Upkie (Figure 3), for which we don't know the true inverse dynamics function and the wheel torque is learned from the ground position and the joint angles.

**Experimental setup**   Learning algorithms are trained on trajectories (a more challenging setting than uniformly spaced data) obtained from multiple system environments. At test time, a new environment is instantiated and the model is adapted from a trajectory of few observations. The resulting adapted model is then used to predict control values for the rest of the trajectory. For the cartpole and the robot arm, the predicted values are used to track a reference trajectory using inverse dynamics control. For Upkie, we could not directly use the predicted controls for actuation, but we compare the open-loop predictions with the executed control law. The target motions are swing-up trajectories for the cartpole and the arm, and a 0.5m displacement for Upkie. Since Upkie is a very unstable system, it is controlled in a 200Hz model predictive control loop (Rawlings, 2000).

**Online adaptive control**   We also investigate a challenging time-varying dynamics setting where the inertial parameters of the system change abruptly at a given time. This scenario is very common in real life and requires the development of control algorithms robust to these changes and fast enough to be adaptive (Åström & Wittenmark, 2013). In our case, we double the mass of the cart in

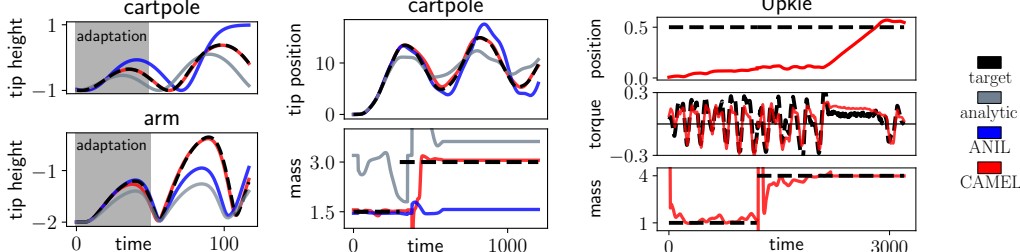

Figure 4: Tracking of a reference trajectory using the learned inverse dynamics controller.
**Left.** 50-shot adaptation. **Center and right.** The model and the controller are adapted online.

the cartpole system, and we quadruple the mass of Upkie's torso. The learning models adapt their task weights online and adjust their control prediction. In an application to parameter identification, we also compute the estimated values of the varying parameter over time.

**Results** The 100-shot adaptation error of the control values is reported in Table 2. The trajectories obtained with inverse dynamics control adapted from 50 shots are plotted in Figure 4 for CAMEL and for the best-performing baseline, ANIL, along with the analytical solution. Only CAMEL adapts well enough to track the target trajectory. The analytic solution underestimates the control as it does not account for friction, resulting in inaccurate tracking. In the adaptive control setting, the variation in the mass of the cart leads to a deviation from the target trajectory but CAMEL is able to adapt quickly to the new environment and identifies the new mass, unlike ANIL. Experimentation on Upkie shows that the computational time of adaptation can be crucial, as we found that the gradient-based adaptation of ANIL and CoDA was too slow to run in the 200Hz model predictive control loop. On the other hand, CAMEL's gradient-free adaptation and interpretability allow it to track and identify changes in system dynamics, and to correctly predict the stabilizing control law.

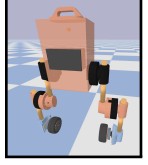

Figure 3: Upkie.

### 5.3 BEYOND CONTEXT-LINEAR SYSTEMS

In order to evaluate our method on general systems with no known parametric structure, we consider the following non-analytical electrostatic problem of the form shown in Example 2. The field is created by a capacitor formed by two electrodes that are not exactly parallel. The variability of the different experiments stems from the misalignment $\delta\varphi \in \mathbb{R}^2$, in angle and position, of the upper electrode. We apply the same methodology as described in Section 5.1. The whole multi-environment learning experiment is repeated several times with varying magnitudes of misalignment, by replacing $\delta\varphi$ with $\varepsilon\,\delta\varphi$ for different values of $\varepsilon \in [0, 1]$. This allows us to move gradually from local perturbations when $\varepsilon \ll 1$ (as in Example 5) to arbitrary variations in the environment.

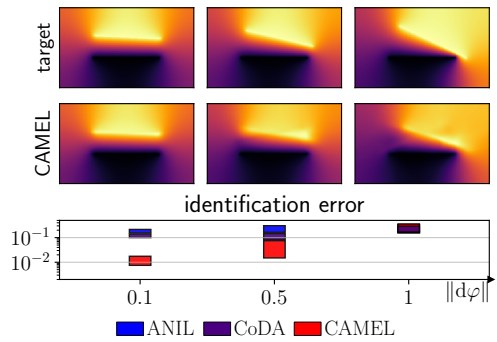

Figure 5: Adaptation and relative identification error for the $\varepsilon$-capacitor, with increasing $\varepsilon$.

**Results** The 40-shot adaptation error for the $\varepsilon$-capacitor is reported in Table 2, with perturbation of full magnitude $\varepsilon = 1$ and with $\varepsilon = 0.1$. We also show the 5-shot adaptation of CAMEL and the best performing baseline, CoDA, for $\varepsilon = 0.2$ in Figure 1. When the system parameters are fully nonlinear, CAMEL and the baselines perform similarly, but CAMEL is much faster. In the second case, CAMEL outperforms them by an order of magnitude and accurately predicts the electrostatic field, whereas CoDA's exhibits lower precision. Predictions and average identification error (with

standard deviations) are plotted as a function of $\varepsilon$ in Figure 5. For small $\varepsilon$, the system parameter perturbation is well identified, enabling a zero-shot adaptation.

Table 2: Average adaptation mean squared error (**left**) and computational time (**right**).

| System | Charges | Capacitor | $\varepsilon$-Capacitor | Cartpole | Arm | Upkie | Training | Adaptation |
|--------|---------|-----------|-------------|----------|-----|-------|----------|------------|
| MAML | 1.6E-1 | N/A | N/A | 1.8E0 | 8.1E-1 | 1.5E-2 | 30 | 10 |
| ANIL | 9.2E-4 | 3.6E-2 | 1.1E-3 | 2.5E-2 | 7.5E-1 | 1.9E-2 | 10 | 3 |
| CoDA | 8.2E-2 | 2.6E-2 | 1.0E-3 | 8.1E-1 | 9.3E-1 | 2.1E-2 | 2 | 8 |
| R2-D2 | 1.2E-4 | 3.1E-4 | 4.2E-4 | 8.5E-3 | 3.5E-1 | 2.3E-2 | 20 | 1 |
| CAMEL | 1.0E-4 | 2.6E-2 | 1.9E-4 | 3.1E-3 | 2.4E-1 | 8.2E-3 | 1 | 1 |

## 6 RELATED WORK

**Multi-task meta-learning** Meta-learning algorithms for multi-task generalization have gained popularity (Hospedales et al., 2021), with the MAML algorithm of Finn et al. (2017) playing a fundamental role in this area. Based on the same principle, the variants ANIL (Raghu et al., 2020) and CAVIA (Zintgraf et al., 2019) have been proposed to mitigate training costs and reduce over-fitting. Interpretability is addressed in the latter work, using a large number of training tasks. In a different line of work, Bertinetto et al. (2019) proposed the R2-D2 architecture where the heads of the network are adapted using the closed-form formula of Ridge regression. The similarities between multi-task representation learning and gradient-based learning are studied in (Wang et al., 2021) from a theoretical point of view, in the limit of a large number of tasks. Unlike our method, the approaches above rely on the assumption that the number of training tasks is large (in few-shot image classification for example, where it can be in the millions (Wang et al., 2021; Hospedales et al., 2021)), while it is typically very limited for physical systems.

**Meta-learning physical systems** Meta-learning has been applied to multi-environment data for physical systems, with a focus on dynamical systems, where the target function is the flow of a differential equation. Recent algorithms include LEADS (Yin et al., 2021), in which the task dependence is additive in the output space and CoDA (Kirchmeyer et al., 2022), where parameter identification is addressed briefly, but under strong assumptions of input linearity. Wang et al. (2022b) propose physical-context-based learning, but context supervision is required for training. From a broader point of view, the interpretability of the statistical model can be imposed by adding physical constraints to the loss function (Raissi et al., 2019).

**Multi-task reinforcement learning** Meta-learning has given rise to a number of fruitful new approaches in the field of reinforcement learning. Sodhani et al. (2021) and Clavera et al. (2019) propose multi-task deep learning algorithms, but no structure is assumed on the dynamics and the learned weights can be interpreted only statistically, in the parameter space of a large black-box neural network. Multi-task learning of inverse dynamics with varying inertial parameters is studied in (Williams et al., 2008) using Gaussian processes, but parameter identification is not addressed.

## 7 CONCLUSION

We introduced CAMEL, a simple multi-task learning algorithm designed for multi-environment learning of physical systems. For general and complex physical systems, we demonstrated that our method performs as well as the state-of-the-art, at a much lower computational cost. Moreover, when the learned system exhibits a linear structure in its physical parameters, our architecture is particularly effective, and enables the identification of these parameters with little supervision, independently of training. The identifiability conditions found in Proposition 1 are not very restrictive, and the effectiveness of the linear identification map is demonstrated in our experiments.

We proposed a particular application in the field of robotics where our data-driven method enables concurrent adaptive control and system identification. We believe that enforcing more physical structure in the meta-model, using for example Lagrangian neural networks (Lutter et al., 2019), can improve its sample efficiency and extend its applicability to more complex robots.

While we focused on classical regression tasks, our framework can be generalized to predict dynamical systems by combining it with a differentiable solver (Chen et al., 2018). Another interesting avenue for future research is the use of active learning, to make the most at out the available training resource and enhance the efficiency of multi-task learning for static and dynamic systems (Wang et al., 2023; Blanke & Lelarge, 2023).

ACKNOWLEDGEMENTS

This work was partially supported by the French government under management of Agence Nationale de la Recherche as part of the "Investissements d'avenir" program, reference ANR19-P3IA-0001 (PRAIRIE 3IA Institute).

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

## A  Proofs

**Lemma 1.** Let $v_1, \ldots, v_N$, and $w_1, \ldots, w_T \in \mathbb{R}^r$, and let $r' \leq r$ and $v'_1, \ldots, v'_N$, and $w'_1, \ldots, w'_T \in \mathbb{R}^{r'}$ be two sets of vector of full rank, satisfying $\forall i, t, w_t^\top v_i = w'^\top_t v'_i$. Then there exist $P, Q \in \mathbb{R}^{r' \times r}$ such that $w'_t = P w_t$ and $v'_i = Q v_i$. Furthermore, $QP^\top = I_{r'}$.

*Proof of Lemma 1.* Denoting by $V \in \mathbb{R}^{N \times r}, V' \in \mathbb{R}^{N \times r'}, W \in \mathbb{R}^{T \times r}$ and $W' \in \mathbb{R}^{T \times r'}$ the matrix representations of the vectors, the scalar equalities $\forall i, t, w_t^\top v_i = w'^\top_t v'_i$ take the matrix form

$$VW^\top = V'W'^\top. \tag{A.1}$$

Since $V'$ is of full rank, the matrix $V'^+ := (V'V'^\top)^{-1}V'^\top \in \mathbb{R}^{r \times N}$ is well defined and is a left inverse of $V'$. Multiplying (A.1) by $V'^+$ yields

$$W' = WP^\top \quad \text{with} \quad P := V'^+ V \in \mathbb{R}^{r' \times r}. \tag{A.2}$$

Similarly,

$$V' = VQ^\top \quad \text{with} \quad Q := W'^+ W \in \mathbb{R}^{r' \times r}. \tag{A.3}$$

Now compute $QP^\top = W'^+ W P^\top = W'^+ W' = I_{r'}$

$\square$

*Proof of Proposition 1.* Applying Lemma 1 to $v'_i := \nu(x^{(i)}), v_i := v(x^{(i)})$, and $w_t := \omega_t, w'_t := \varphi_t$ yields the stated result.
$\square$

The case where $c, \kappa \neq 0$ can be handled as follows. We augment $\varphi$ and $\nu$, and $\omega$ and $v$ with an additional dimension, with the last components of $\varphi$ and $\omega$ equal to 1 and the last components of $\nu$ and $v$ equal to $\kappa$ and $c$ respectively. The augmented vectors satisfy the assumptions of Proposition 1 provided the augmented $v'_i$ and $w'_t$ span $\mathbb{R}^{n+1}$. The proposition then applies, and implies that the physical parameters $\varphi_t$ can be recovered with an affine transform. This case is tackled experimentally in the capacitor experiment (Section 5.3), where $\kappa \neq 0$ *a fortiori* since the electrostatic field is linearized around a nonzero value. The physical parameters are identified using an affine regression.

## B  Experimental details

### B.1  Architectures

All neural networks are trained with the ADAM optimizer Kingma & Ba (2015). For CoDA, we set $d_\xi = r$, chosen according to the system learned. For all the baselines, the adaptation minimization problem (2.5) is optimized with at least 10 gradient steps, until convergence.
For training, the number of inner gradient steps of MAML and ANIL is chosen to be 1, to reduce the computational time. We have also experimented with larger numbers of inner gradient steps. This improved the stability of training, but at the cost of greater training time.

### B.2  Systems

We provide further details about the physical systems on which the experiments of Section 5 are performed.

#### B.2.1  Point charges

The $n$ charges are placed at fixed locations in the plane at fixed location. The training inputs are located in $\Omega = [-1, 1] \times [0, 1]$ which is discretized into a $20 \times 20$ grid and the ground truth potential field is computed using Coulomb's law.
The training data is generated by changing each charge's value in $\{1, \ldots, 5\}^n$, hence $T = 5^n$. We have experimented on different settings with various numbers of charges, and various locations. In Section 5.1, a dipolar configuration is investigated, where $n = 3$, and one of the charges is far away on the left and two other charges of opposite sign are located near $x_2 = 0$. Gaussian noise of size $\sigma = 0.1$ is added to the field values revealed to the learner in the test dataset.
The system is learned with a neural network of 4 hidden layers of width 16, with the last layer of size $r = n$.
For evaluation, the test data is generated with random charges drawn from a uniform distribution in $[1, \ldots, 5]^n$ and the data points are drawn uniformly in $\Omega$

### B.2.2 CAPACITOR

The space is discretized into a $200 \times 300$ grid. The training environments are generated with 10 values of the physical context $\varphi := (\alpha, \eta) \in [0, 0.5] \times [-0.5, 0.5]$ containing the angular and the positional perturbation of the second plate, drawn uniformly. The ground truth electrostatic field is computed with the Poisson equation solver of Zaman (2022). For evaluation, 5 new environments are drawn with the same distribution.

The system is learned with a neural network of 4 hidden layers of width 64, with the last layer of size $r = n + 1 = 3$.

### B.2.3 CARTPOLE AND ARM

We have implemented the manipulator equations for the cartpole and the arm (or acrobot), following Tedrake (2022), and have added friction. The training data is generated by actuating the robots with sinusoidal inputs, with for each environment 8 trajectories of 200 points and random initial conditions and periods. At test time, the trajectories are generated with sinusoidal inputs for evalutation, and with swing-up inputs for trajectory tracking.

**Cartpole** The pole's length is set to 1, the varying physical parameters are the masses of the cart and of the pole: $\varphi_t \in \{1, 2\} \times \{0.2, 0.5\}$, so $T = 4$. For evaluation, the masses are drawn uniformly around $(2, 0.3)$, with an amplitude of $(1, 0.2)$. The system is learned with a neural network of 3 hidden layers of width 16, with the last layer of size $r = n + 2 = 4$.

**Arm** The arm's length are set to 1, the varying physical parameters are the inertia and the mass of the second arm: $\varphi_t \in \{0.25, 0.3, 0.4\} \times \{0.9, 1.0, 1.3\}$, so $T = 9$. For evaluation, the inertial parameters are drawn uniformly around $(0.5, 1)$, with an amplitude of $(0.2, 0.3)$. The system is learned with a neural network of 4 hidden layers of width 64, with the last layer of size $r = n + 2 = 4$.

### B.2.4 UPKIE

Information about the open-source robot Upkie can be found at `https://github.com/tasts-robots/upkie`.

We trained the meta-learning algorithm on balancing trajectories of 1000 observations, with 10 different values for Upkie's torso, ranging from 0.5 to 10 kilograms. For evaluation, the mass is sampled in the same interval.

The system is learned with a neural network of 4 hidden layers of width 64, with the last layer of size $r = n + 2 = 3$.

### B.3 INVERSE DYNAMICS CONTROL

The Euler-Lagrange formulation for the rigid body dynamics has the form

$$M(q)\ddot{q} + C(q, \dot{q})\dot{q} + g(q) = Bu, \tag{B.1}$$

where $q$ is the generalized coordinate vector, $M$ is the mass matrix, $C$ is the Coriolis force matrix, $g(q)$ is the gravity vector and the matrix $B$ maps the input $u$ into generalized forces (Tedrake, 2022).

Inverse dynamics control is a nonlinear control technique that aims at computing the control inputs of a system given a target trajectory $\{\bar{q}(s)\}$ Spong et al. (2020). Using a model $\hat{\text{ID}}$ for the inverse dynamics equation (B.1), the feedforward predicted control signal $\hat{u} = \hat{\text{ID}}(\bar{q}, \dot{\bar{q}}, \ddot{\bar{q}})$. These feedforward control values can then be combined with a low gain feedback controller to ensure stability, as

$$u = \hat{u} + K(\bar{q} - q) + K'(\dot{\bar{q}} - \dot{q}). \tag{B.2}$$

For the cartpole, we used $K = K' = 0.5$. For the robot arm, we used $K = K' = 1$.

### B.4 ADAPTIVE CONTROL

In a time-varying dynamics scenario, CAMEL can be used for adaptive control and system identification. Given a target trajectory, the task-agnostic component $v$ of the model predictions can be computed offline. In the control loop, the task-specific component $\omega$ is updated with the online least squares formula. The control loop is summarized in Algorithm 2, where we have assumed $c = 0$ for simplicity. The estimated inertial parameters are deduced from the task-specific weights with the identification matrix (4.4).

---

**Algorithm 2** Adaptive trajectory tracking

---

    **input** trained feature map $v(x)$, target trajectory $s \mapsto \bar{q}_s$
    **Offline control**
    **for** timestep $0 \leq s \leq H-1$ **do**
        compute $\bar{x}_s = (\bar{q}_s, \dot{\bar{q}}_s, \ddot{\bar{q}}_s)$
        compute features $\bar{v}_s := v(\bar{x}_s)$
    **end for**
    **Control loop**
    Initialize $M_0 = I_r, \quad \omega_0 = (0, \ldots, 0)$
    **for** time step $1 \leq s \leq H$ **do**
        compute $\hat{u}_s = \omega_s^\top \bar{v}_s$
        compute $e_s = q_s - \bar{q}_s$
        play $u_s := \hat{u}_s + K e_s$
        observe $q_{s+1}, \dot{q}_{s+1}$
        compute $v_s := v(x_s)$
        update $M_{s+1} = M_s - \frac{M_s v_s (M_s v_s)^\top}{1 + v_s^\top M_s v_s}$
        update $\omega_{s+1} = \omega_s - (v_s^\top \omega_s - u_s) M_{s+1} v_s$
    **end for**

---

## B.5 ADDITIONAL NUMERICAL RESULTS

We provide details concerning Table 2.

**Computational time**   For the computational times of Table 2, we arbitrarily chose the shortest time as the time unit, for a clearer comparison among the baselines. The computational times were measured and averaged over each experiment, with equal numbers of batch sizes and gradient steps across the different architectures. For training, the time was divided by the number of gradient steps.

Table 3: Adaptation performances with standard deviations.

| System | Charges, 30 trials | | Capacitor, 5 trials | |
|---|---|---|---|---|
| | 3-shot | 10-shot | 5-shot | 40-shot |
| MAML | 4.1E-0 ± 2E-0 | 1.6E-1 ± 5E-2 | N/A | N/A |
| ANIL | 3.5E0 ± 5E-1 | 9.2E-4 ± 5E-4 | 4.4E-2 ± 2E-2 | 3.6E-2 ± 1E-2 |
| CoDA | 1.0E-1 ± 9E-2 | 8.2E-2 ± 3E-2 | 4.7E-2 ± 5E-5 | 2.6E-2 ± 1E-2 |
| CAMEL | 2.0E-4 ± 1E-4 | 1.0E-4 ± 5E-5 | 3.6E-2 ± 2E-2 | 2.6E-2 ± 1E-2 |
| $\varphi$-CAMEL | 3.0E-3 | | 6.5E-2 | |

| System | $\varepsilon$-Capacitor, $\varepsilon = 0.1$, 5 trials | |
|---|---|---|
| | 3-shot | 30-shot |
| MAML | N/A | N/A |
| ANIL | 1.1E-3 ± 5E-5 | 1.1E-3 ± 5E-5 |
| CoDA | 1.2E-3 ± 5E-4 | 1.0E-3 ± 5E-4 |
| CAMEL | 4.2E-4 ± 1E-4 | 1.9E-4 ± 2E-5 |
| $\varphi$-CAMEL | 1.9E-4 | |

| System | Cartpole, 50 trials | | Arm, 50 trials | |
|---|---|---|---|---|
| | 50-shot | 100-shot | 50-shot | 100-shot |
| MAML | 4.3E0 ± 7E-1 | 3.5E0 ± 6E-1 | 1.0E0 ± 1E-1 | 8.1E-1 ± 5E-2 |
| ANIL | 3.8E-1 ± 1E-1 | 2.5E-2 ± 9E-2 | 8.5E-1 ± 1E-1 | 7.5E-1 ± 4E-2 |
| CoDA | 3.8E-1 ± 9E-3 | 8.1E-1 ± 1E-1 | 9.5E-1 ± 9E-2 | 9.3E-1 ± 6E-2 |
| CAMEL | 4.8E-2 ± 1E-2 | 3.1E-3 ± 5E-4 | 3.1E-1 ± 5E-2 | 2.4E-1 ± 1E-2 |

| System | Upkie, 15 trials |
|---|---|
| MAML | 1.5E-2 ± 7E-3 |
| ANIL | 1.9E-2 ± 6E-3 |
| CoDA | 2.1E-2 ± 3E-3 |
| CAMEL | 8.2E-3 ± 5E-3 |

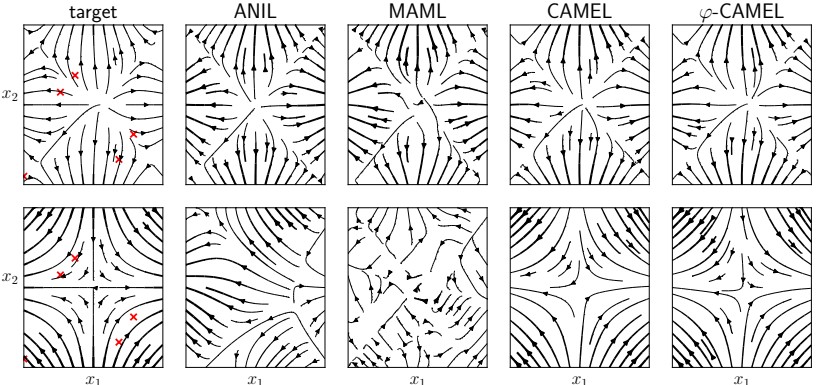

Figure 6: 5-shot adaptation for the 4 point charge system. **Top.** The four charges are positive, as in the training meta-dataset. **Bottom** Two of the four charges are negative.

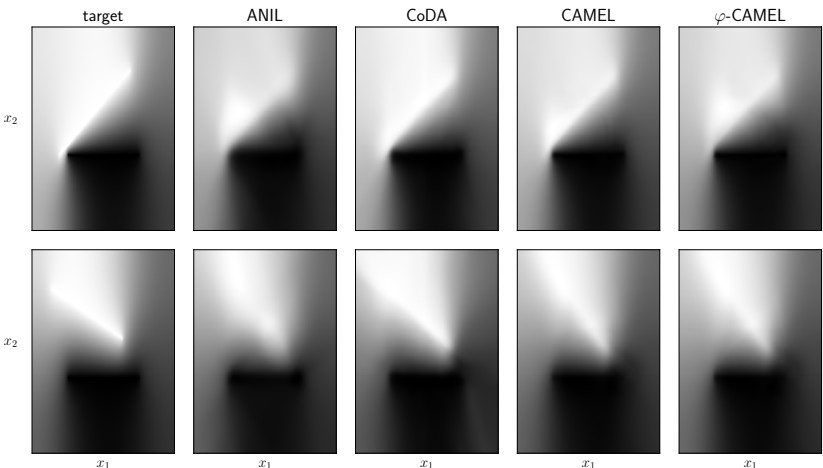

Figure 7: Capacitor, 40-shot adaptation.

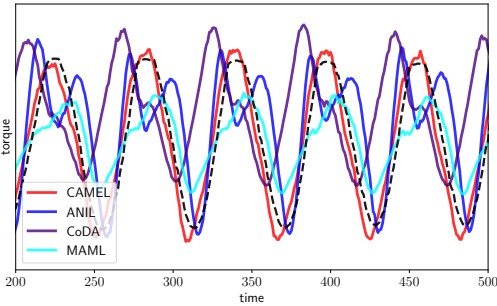

Figure 8: Upkie torque prediction, 100-shot adaptation.

## B.6 ZERO-SHOT ADAPTATION AND SCIENTIFIC DISCOVERY

**Zero-shot adaptation**    Looking at the problem from another angle, Proposition 1 also shows that $\omega$ can be estimated linearly as a function of $\varphi$, at least when $r = n$ (which ensures that $P$ is nonsingular). Computing an estimator of $\omega$ as a function of $\varphi$ with the inverse regression to (4.4) enables

a zero-shot (or physical parameter-induced) adaptation scenario: when an estimate of the physical parameters of the new environment is known a priori, a value for the model weights can be inferred. We call this adaptation method $\varphi$-CAMEL.

In a data-driven approach, training CAMEL offers not only the ability to adapt to a small number of observations, but also to predict the system without any data for arbitrary values of the its parameters. We believe that the 0-shot adaptation algorithm $\varphi$-CAMEL can be used in the process of scientific discovery. In many cases, the experimenter has the knowledge of (or knows an estimate of) the physical quantities varying across experimental conditions, while not knowing accurately the system itself. Then, $\varphi$-CAMEL can be used to infer the target function for chosen values of the physical parameters $\varphi$ independently of the values observed for training.

Of course, the predictions of $\varphi$-CAMEL are good only if the estimator $\hat{\varphi}$ of (4.4) is good, implying a sufficient number of training tasks and an effective training of CAMEL. For nonlinear physical contexts, the values of $\varphi$ that are investigated should be close to the reference value $\varphi_0$ so that (4.2) holds.

We further illustrate on the toy example of $n = 4$ point charges, for which the experimenter could observe experiments with positive charges. Figure 6 shows the predictions after 5-shot adaptation of the different meta-models, along with the zero-shot adaptation of $\varphi$-CAMEL. We can see that only CAMEL and $\varphi$-CAMEL adapt well to negative charges. In particular, the zero-shot adaptation of $\varphi$-CAMEL enables estimating the system in an experiment whose numerical values are completely different from the training dataset, thanks to the structure of the model and of the equations in this case (since they are known to be linear in the charges). Importantly, evaluating $\varphi$-CAMEL for different values of $\varphi$ is not costly, since the identification map is already computed using the training data.

We could imagine that this scenario might enable discovering new properties of complex physical systems as by exploring the space of physical parameters, in a data-driven fashion. Regarding the simple example of Figure 6, knowing the form of the electrostatic field in this quadrupole setting underlies the understanding of Penning's ion trap Kretzschmar (1991).

