# OpenReview forum: "Interpretable Meta-Learning of Physical Systems"
_ICLR.cc/2024/Conference — ICLR 2024 poster_

### Official Review · Reviewer_X11Z · 2023-10-31

**Soundness:** 3 good
**Presentation:** 2 fair
**Contribution:** 2 fair
**Rating:** 5
**Confidence:** 4

**Summary:**

The paper proposes an approach CAMEL for the adaptation to new tasks by first learning a shared neural network feature extractor together with separate linear heads for each training task, then at test time perform least square regression to compute the linear head weights given a new task’s labelled data. The authors show this formulation can allow physical parameters identification for linearly parameterized systems under certain conditions. Experimentally, the authors show that their approach can outperform meta-learning baseline methods (MAML, ANIL, CODA) in terms of generalization, computational speed, and interpretability.

**Strengths:**

The interpretability and physical parameter identification aspects this paper focuses on are important and often overlooked by existing meta-learning methods.

**Weaknesses:**

- **The proposed method lacks novelty and are not situated in the right context**. The authors claim to propose a novel “model agnostic meta-learning architecture." However, the proposed approach is not model agnostic (because it requires specifying the width of the last layer dimension) and more importantly is not novel and not situated in the most relevant context.
    - **Novelty**: It could be argued that the proposed method is not meta-learning but applying multi-task representation learning in a learning-to-learn problem. A more general version of the authors’ proposed approach have been theoretically studied in terms of its generalization performance by Maurer et al  in 2016 (page 7, section 2.2 Bounding the Excess Risk for Learning-to-learn). More recently, Wang et al 2021 have proposed almost exactly the same approach by first multi-task learning a shared feature extractor with linear heads and then performing last linear layer fine-tuning over a new task’s data at test time (section 3.4 Fine-tuning for Test Task Adaption). Besides, Wang et al has also studied the computational time savings of this MTL approach compared to last-layer MTL methods. Thus it is not correct for the authors to claim their methods as a novel approach.
    - **Context**: In the background section, the authors situate their methods among Gradient-based Meta-learning methods. However as the authors have shown in Table 1, CAMEL doesn’t perform gradient adaptation, so it’s not a gradient-based method. Instead, because CAMEL adapts only the last layer of the learned model, a more appropriate way to situate the current method is to compare it among last-layer meta-learning methods, which includes not only ANIL (which the authors have compared against) but more importantly MetaOptNet (Lee et al 2019) and R2D2 (Bertinetto et al., 2019). Here both MetaOptNet and R2D2 solve the last linear layer exactly (using a convex solver or closed-form expression) and use implicit function theorem to propagate the gradients to the earlier feature extractors. Because this paper focuses on the regression task, the authors should compare additionally against R2D2 in the experiments. (One can think of R2D2 roughly as ANIL where the last layer adaptation are solved exactly instead of using a fixed number of gradient descent steps.)

    Maurer, A., Pontil, M., and Romera-Paredes, B. The beneﬁt of multitask representation learning. The Journal of Machine Learning Research, 2016.

    Wang, H., Zhao, H., & Li, B. Bridging multi-task learning and meta-learning: Towards efficient training and effective adaptation. In International conference on machine learning, 2021

    Bertinetto, L., Henriques, J. F., Torr, P., and Vedaldi, A. Meta-learning with differentiable closed-form solvers. In International Conference on Learning Representations, 2019.

    Lee, K., Maji, S., Ravichandran, A., and Soatto, S. Metalearning with differentiable convex optimization. In CVPR, 2019

- **Notation needs to be more consistent and clearer.** For example, both $f_t(x; \pi)$ (page 3) and $f_t(x)$ (page 4) are used. Besides, the ground truth function is denoted $v_{\star}$. However, it has $n$ output dimensions, while the function $v$ has $r$ output dimensions.

- **Proposition 1** are stated without all the assumptions of the relationships between $r$, $n$, $T$, and $N$ (some of these relationships are scattered in earlier parts of the paper), making the statement difficult to parse. Besides, the assumption that $c = c_{\star} = 0$ to me seems a bit restrictive as the ground truth $c_{\star}$ are inherent to the physical process and could be nonzero. However, I’m not sure if the authors can make the same type of guarantees for the cases when these values are non-zero (and possibly unequal).
- In the computational benefit section, the authors mention “adaptation at test time … is guaranteed to converge if the number of samples is greater than $r$.” This statement is a bit confusing, as the optimization is an ordinary least square which can be solved analytically without a notion of convergence. Maybe the authors mean to say when the number of samples is greater than $r$, the ordinary least squares is likely to be fully specified, thus admitting one unique solution. If this is the case, the authors should also discuss how to pick the solution when the ordinary least squares is underspecified.

**Questions:**

- How many steps of gradient adaption is used for ANIL? It is important to note that one can apply more steps of adaptation during evaluation than training as it can potentially improve ANIL's performance.
- Can the authors explain for example 1 and 4, what is the learning goal here? I saw the authors mention $x=(q, \dot q, \ddot q)$ and $y=u$, but can the authors explain why do we want to predict $u$ (which I'm assuming is a function of time) in the first place?

---

> ### Author Response · Authors · 2023-11-15
> **We clarify the novelty of our work and we better situate it in the context of multi-task meta-learning**
>
> We thank you for your time and dedication to this review.  We address each of your questions below.
>
> We first address the main concerns regarding the novelty and the context of our approach. The rest of the questions is addressed in a second message, along with an additional experiment that we conducted for further comparison with existing work.
>
> ## Novelty of our method
>
> We agree that the representation learning architecture with a shared feature map and task-specific linear weights is not novel. We will make it clearer by better referencing (Caruana, 1997) and adding the references below. Our contribution demonstrates the effectiveness of this representation learning architecture in a challenging setting with very few tasks, each tasks containing a large number of points. This framework is motivated by the interpretable learning of physical systems.
>
> In (Wang *et al.*, 2021), representation-based multi-task learning is compared with gradient-based meta-learning algorithms such as MAML and ANIL. As in our paper, both architectures are presented as methods for generalizing to new tasks, by minimizing a regularized multi-task loss function. This work theoretically demonstrates that representation learning can achieve performances comparable to those of meta-learning algorithms, within the limit of large neural networks. These theoretical conclusions are in line with the practical conclusions of our paper in the context of physical systems, which however have a different framework and scope.
>
> While the scope of these works is indeed similar to ours, we would like to point out some important differences. Our approach lies in the context of learning physical systems, where data collection is costly and computational resources can be limited. Typically, the number of training tasks is small (the number of tasks $T$ can be smaller than $4$ for example, as in our experiments), and the neural networks used are limited in width and depth to avoid overfitting. Consequently, the theoretical guarantees of (Maurer *et al.*, 2016), Theorem 2, and (Wang *et al.*, 2021), Theorem 1, where the number of tasks and the network depth tend to infinity, do not apply to our use cases. We also point out that we focus on regression tasks, while the experiments carried out in (Wang *et al.*, 2021) focus on classification. Furthermore, in the few-shot image classification experiments in the latter work, $T$ is very large ( $\binom{5}{64} \simeq 7.10^6$) and the number of training points per task is very small (typically smaller than 10), as opposed to our setting where $T$ is small and the number of training points $N_t$ is not restricted ($N_t$ of order 10000 for the training grid in the capacitor experiment).
>
> Unlike image classification, we show that the linear heads in the representation learning architecture can exploit the structure of the problem to provide a low-cost, high-performance algorithm for learning an interpretable physical model. In image classification, the algorithm has access to a very large database, which compensates for the lack of structure in the learning problem. In our submission, we show that, in a very different framework with less data and less compute, complex functions can still be learned efficiently with an interpretable model by exploiting the linear structure of the problem. Hence, like (Wang *et al.*, 2021), we demonstrate that representation learning can match more complex meta-learning architectures, but our study is carried out in a different setting, with different motivations and scope, and with the crucial benefit of identifying the underlying physical parameters.
>
> ## Context
>
> We thank you for the references and we will make sure to add them to the text body, and to better situate the context of our meta-learning algorithm.
>
> The R2D2 algorithm indeed has in common with CAMEL the adaptation of the network head for each task. Using the formalism of our paper, R2D2 computes task specific weights $w_t = A(\pi, D_t)$ with $A(\pi, D_t)$ solving a Ridge regression of the $N_t$ points of $D_t$, for each training step and for each task.
>
> R2D2 should perform well for multi-task learning thanks to the complete resolution of the regression at each learning task. However, implementing it in our setting where $N_t$ is very large is not straightforward, as Ridge regression requires the inversion of a quadratic-sized system into the number of the features or of the regression points (which can be in the tens of thousands in our experiments as we mentioned above), hence adding a substantial amount of gradient computations. We have implemented this method and compared it to CAMEL in our experiments, and we report our results in a separate message.

---

> > ### Author Response · Authors · 2023-11-15
> > **Additional experiment and answers to the questions**
> >
> > # Additional experiments
> >
> > We propose an additional experiment to address the concern regarding the comparison of CAMEL to another method that adapts only the heads of the network.
> >
> > We have implemented the R2D2 multi-task learning algorithm and have tested it on two static system from our experiments. For the Ridge regression, we chose a regularizer of $\lambda = 10^{-3}$ and did not encounter any singularity error. We used the same neural networks for both approaches. The results are reported in the following table.
> >
> > |experiment|test error|computational time|
> > |---|---|---|
> > |CAMEL on Charges|1.0e-4|1|
> > |R2D2 on Charges|1.2e-4|2|
> > |CAMEL on Capacitor|1.9e-4|1|
> > |R2D2 on Capacitor|4.2e-4|70|
> >
> > For both systems,  R2D2's generalization performance matches that of CAMEL. This is expected because both algorithms learn a shared feature representation and adapt linear heads for each task.
> >
> > However, the computational cost of R2D2 is much higher, owing to the higher cost of computing Ridge regression and its gradient with respect to model parameters. This additional cost becomes prohibitive when the number of gradient operations becomes very large, in particular for the Capacitor $N_t = 60000$. Whether using the Woodbury formula or not, the training time between CAMEL and R2D2 goes from minutes to hours.
> >
> > This result underlines the difference between the two methods. In image classification, network heads cannot be kept in memory for all training tasks, so they are computed at each episode, which induces an additional gradient cost, but which is not limiting because the number of $N_t$ points is small.
> > For the physical systems we're concerned with here, the number of training tasks is small, so training a multi-task network while keeping the heads associated with the different tasks is possible. This is even preferable, since the number of points for each task is large, so the number of gradients to be calculated should be as small as possible and costly operations such as those in MAML and R2D2 become prohibitive.
> >
> >
> > We now answer the other questions. We thank you for pointing out notation and clarity issues; we will fix them in an updated version of the submission.
> >
> > ## Proposition 1
> >
> > Regarding the case where $c, c_\star \neq 0$, it can be handled by augmenting  $\varphi$ and $v_{\star}$, and $w$ and $v_\star$ with an additional dimension, with the corresponding component of $v$ and $v_\star$ equal to $1$. The conclusion of Proposition 1 then applies wit the assumption of $\{v_\star(x^{(i)})\}$ spanning $\mathbb{R}^{n+1}$. Hence, the augmented physical parameters can be recovered up to a linear transform, meaning that $\varphi$ can be recovered with an affine transform. We have tackled this case $c, c_\star \neq 0$ experimentally in the capacitor experiment, where $c_\star$ is non zero a fortiori because the electrostatic field is linearized around a nonzero value.
> >
> > ## Questions
> >
> > ``How many steps of gradient adaption is used for ANIL? ``
> >
> > In the training inner loop, we used only one adaptation step. Adding more might make the model more expressive, but the computational cost would be even larger. At inference time, we have used a number of steps varying between 10 and 50, depending on the experiment. We chose the number of adaptation steps (and the stepsize) carefully so that the adaptation loss converged to is minimal value as in (2.5).
> >
> > ``Can the authors explain for example 1 and 4, what is the learning goal here?``
> >
> > We indeed want to predict the action $u$ as a function of the state, velocity and accelerations. The goal here is to track a reference trajectory, hence we are given the target state and its derivatives. Since the problem is "what control law $u$ should be input to the system to reach the desired trajectory?", a model that predicts $u$ as a function of the trajectory information can be used in combination with proportional controllers to design a feedback controller for trajectory tracking, which is called inverse dynamics control. This type of control is very common in robotics. More information can be found in Appendix B.2 of our submission, and in (Spong *et al.*, 2020).
> >
> > ## References
> >
> > Rich Caruana. Multitask learning. Machine learning, 28:41–75, 1997.
> > Wang, H., Zhao, H., & Li, B. Bridging multi-task learning and meta-learning: Towards efficient training and effective adaptation. In International conference on machine learning, 2021
> > Bertinetto, L., Henriques, J. F., Torr, P., and Vedaldi, A. Meta-learning with differentiable closed-form solvers. In International Conference on Learning Representations, 2019.
> > Mark W Spong, Seth Hutchinson, and Mathukumalli Vidyasagar. Robot modeling and control. John
> > Wiley & Sons, 2020
> > Bertinetto, L., Henriques, J. F., Torr, P., and Vedaldi, A. Meta-learning with differentiable closed-form solvers. In International Conference on Learning Representations, 2019.

---

> > > ### Comment · Reviewer_3DG3 · 2023-11-16
> > > **My quick thoughts here**
> > >
> > > I agree with the original review that the novelty of the work is important for the paper to discuss, especially w.r.t. Wang et al. and R2D2 as they also consider linear heads. The original submission is clearly lacking without references to these, and I also agree with this reviewer's comment that the approach is /not/ model agnostic.
> > >
> > > The author's response here seems reasonable and would address these issues if included in the final version of the paper. My initial impression is to agree with the authors that their methodology and experimental settings are still sufficiently different that there is still value in their contribution. And, the new R2D2 comparison with the timing results is interesting and helps connect to these topics.
> > >
> > > As I stated in my review thread, I still think the paper is lacking an experimental comparison to existing published results --- if the approach is computationally faster than R2D2 in the settings above, maybe it's also worth comparing to R2D2 (or Wang et al./MetaOptNet) on their experimental settings too.

---

> > > ### Comment · Reviewer_rUF2 · 2023-11-20
> > > **Comments from reviewer rUF2**
> > >
> > > I appreciate the points brought up by Reviewer X11Z, especially the comparison to Wang et al. and Bertinetto et al. In my opinion, the author's response takes these points into account and responds appropriately. The inclusion of an experimental comparison to these works will strengthen the article and improve its novelty. When assessing novelty, I also believe that the application case to physical modelling should be considered, as Wang et al. and Bertinetto et al. both consider image classification tasks.

---

> > > > ### Comment · Reviewer_X11Z · 2023-11-22
> > > > **Response to author rebuttal**
> > > >
> > > > I would like to thank the authors and Reviewers 3DG3 and rUF2 for their response. I provide response to the authors’ points below:
> > > >
> > > > **Novelty**. I acknowledge the authors’ discussion of the novelty of their method. If I understand correctly, the authors’ stated contributions in the paper are 1) proposing CAMEL, 2) showing CAMEL is computationally more efficient than other meta-learning approaches (which require more gradient computation), 3) applying CAMEL to learning physical systems (which is not often discussed in meta-learning), 4) showing CAMEL can identify physical parameters for linearly parameterized systems, thus providing additional interpretability; 5) showing CAMEL is competitive with other approaches in terms of performance on the physics tasks. I do not question the novelty of points 3) - 5), but I still strongly believe that, given that Wang et al’s method (description in 3.4 Fine-tuning for Test Task Adaption in their paper) is exactly the same as CAMEL (except it’s experimentally tested on classification), it’s not longer correct to claim any novelty by saying CAMEL “is a new meta-learning architecture.” Wang et al didn’t give a new name to the method they studied and simply called it MTL (multi-task learning), so I believe the current paper’s narrative of giving this old method a new name and claiming it as a new method is a bit unfair. Besides, the computation cost is also something that Wang et al has investigated. I can accept that applying MTL (CAMEL) to the physics setting can still be deemed novel for interpretability purposes, but I would like to see the authors give a concrete explanation as to whether/how they plan to change their narrative in the introduction of the method CAMEL.
> > > >
> > > > **Comparison against R2D2.** I find the authors’ response on the R2D2 comparison satisfactory. I would strongly suggest the authors include the R2D2 experiment comparison in the main paper (if not comparing against it on every experiment). However, I believe the authors should emphasize more clearly their focus on a few number of training tasks and a large of number of examples. In addition, the fact that one has $N_t=60000$ for a single task doesn’t necessarily mean one needs to adapt R2D2 to all of them in the inner loop during the meta-training stage. Instead, one can simply sample a small number of them as the support set and some others as the query set. However, during meta-test time, we can still perform last layer learning over all the available labeled data. (This approach is similar to the way we train MAML, we can only train with 1 or 5 inner steps during meta-training but perform $>10$ steps adapation during meta-testing.) Regardless of this comment above, I accept the authors’ observation that R2D2 will take longer to meta-train, as a similar (not the same) comparison is performed by Wang et al in Table 4 between MTL and MetaOptNet.
> > > >
> > > > **Proposition 1 and Questions**. I accept the authors’ explanation and answers. I recommend the authors provide the complete theorem statement and proof details for $c, c_{\star} \neq 0$ case in their revision.
> > > >
> > > > **Summary**. Overall, I’m willing to consider increasing my score from 3 to 5 if the authors can convince me that they will change the paper’s narrative on CAMEL (by acknowledging its lack of “methodology novelty”) and sufficiently acknowledge prior work (instead of only finding details on how CAMEL differs from them).

---

> > > > > ### Author Response · Authors · 2023-11-22
> > > > > **Thank you for your response, we uploaded a revised version taking it into account**
> > > > >
> > > > > We thank you for you response.
> > > > >
> > > > > We have uploaded an updated version of the manuscript, with the major additions higlighted in blue and the deletions highlighted in red.
> > > > >
> > > > > In particular, this updated version changes the narrative on CAMEL following your remarks: we better reference prior work on multi-task meta-learning and we clarify the novelty of our approach in the context of learning physical systems.
> > > > >
> > > > > More precisely,  rather than claiming that we introduce a new architecture, we present our contribution as an adaptation of the Multi-Task Learning algorithm of (Wang *et al.*, 2021)  that is motivated by structural considerations: this architecture incorporates structure and is hence well-suited for learning physical systems efficiently in an interpretable way.
> > > > >
> > > > > The comparison of our work to other approaches, with an explanation of how the setting of image classification and that of physical systems differ, is provided in the Related work section.
> > > > >
> > > > > We have also added in the main body the comparison against R2-D2, which we have run on all the systems presented in our paper.
> > > > >
> > > > > **Terminology**
> > > > > We would like to draw your attention to a terminology matter. In (Wang *et al.*, 2021), the multi-task representation learning algorithm is called simply "Multi-Task Learning" (MTL). However, we believe that this expression may also refer to the more general framework of multi-task learning as described in (Caruana *et al.*, 1997). For maximum clarity when making comparisons with other approaches, we have opted to retain the concise name "CAMEL", making it clear that this is another name for the same architecture, adapted to our physical system learning framework. The architecture is first introduced as multi-task learning (MTL) in the introduction and in Definition 1, and then the name CAMEL is introduced right after Definition 1 (page 4) in our updated manuscript.
> > > > >
> > > > > Please us know if you have any further questions.
> > > > >
> > > > > ## References
> > > > >
> > > > > Rich Caruana. Multitask learning. Machine learning, 28:41–75, 1997.
> > > > >
> > > > > Wang, H., Zhao, H., & Li, B. Bridging multi-task learning and meta-learning: Towards efficient training and effective adaptation. In International conference on machine learning, 2021

---

> > > > > > ### Comment · Reviewer_X11Z · 2023-11-22
> > > > > > **Follow-up response**
> > > > > >
> > > > > > I would like to thank the authors in their response and in changing the narrative of the paper. I also believe that it is acceptable to use CAMEL to more compactly and unambiguously refer to the practice of learning a linear head over a multi-task learned representation on a new task. Thus I have increased my rating from 3 to 5.

---

> ### Author Response · Authors · 2023-11-19
> **As the deadline approaches, are there any other questions that we can address?**
>
> We would like to thank you once again for your insightful comments. We did our best to answer all of the questions in detail, especially the main concern regarding the novelty of our approach, for which  we also provided an additional experiment.
>
> As the deadline for the author-reviewer discussion (Wednesday, November 22) approaches, please let us know if there are any additional questions or concerns that we can address.

---

### Official Review · Reviewer_3DG3 · 2023-11-01

**Soundness:** 2 fair
**Presentation:** 4 excellent
**Contribution:** 3 good
**Rating:** 6
**Confidence:** 3

**Summary:**

The paper investigates meta-learning physical systems via
associating every task with a linear embedding/task weight
w that impacts the predictions: this is shown in equation 3.2
where the prediction involves a global prediction
followed by a modification from the linear part.
At test-time, the task weight of a new task is found
by gradient descent on the task loss.
Section 4 goes on to describe many settings where it is
reasonable to have this linear part of the model,
and prop 1 shows that the task weights can recover
the true dynamics of the system, which leads to the
an interpretability of what the task weights are.
The experimental result show a number of settings
from physics and reinforcement learning where
the method is able to learn better meta-dynamics models
than MAML, ANIL, and CoDA.

**Strengths:**

1. This is a reasonable model when there is some reason to believe
   the true model follows the formulation of equation 3.2.
   It could be interesting to further speculate about the
   representation capacity or universality of this model.
   (Section 4 is perhaps a good start to this)
2. The experimental results clearly demonstrate the
   capability of the model, especially in how well CAMEL
   is able to attain a low identification error in Fig 2.
3. CoDA is a great baseline method to compare against,
   in addition to MAML/ANIL, and Table 1 clearly summarizes
   the modeling differences.

**Weaknesses:**

1. Every experimental meta-learning setting in this paper
   seems to be newt
   This would have been helpful to help better-ground the results
   in the community, as otherwise it is difficult
   to understand if the performance improvements are coming
   from improper implementations of other methods
   or from a fundamental modeling advancement.
   For example, MAML with a training step size of 30 and
   adaptation size of 10 in Table 2 is very arbitrary and
   unlikely to be the best MAML hyper-parameters here.

   The paper would be significantly more convincing
   if the CAMEL model was evaluated in the exact experimental settings
   from CoDA or [Wang et al., NeurIPS 2022](https://arxiv.org/abs/2102.10271),
   which also meta-learn physical systems with varying amounts of
   contextual information.
2. There is a significant amount of omitted related work on
   meta-reinforcement learning that is also adapting controlled
   MDP dynamics and policies to multi-task RL settings,
   e.g., [Nagabandi et al., ICLR 2019](https://arxiv.org/abs/1803.11347)
   and the citation graph around it.
3. The term "interpretable meta-learning" in the title and
   throughout the rest of the paper is a big statement without
   the right qualifications. The paper gets to defining what
   interpretability means here in proposition 1 of recovering
   true parameters when they exist and fit into the modeling
   structure they have set up, but I this is not what I had
   in mind when I first saw the term "interpretability".
   It also seems like there is still a large amount of
   uninterpretability in the system, as the model in
   equation 3.2 still has 2 neural networks.

**Questions:**

Overall I think this is a well-executed paper with some
good modeling ideas and experimental insights.
I gave the paper a weak reject due to the experimental settings
not being directly comparable to the related work and the
baselines not being tuned as much as they could have been,
and I am especially open to re-evaluating this assessment
after hearing the author's perspective on this.

---

> ### Author Response · Authors · 2023-11-15
> **About the experimental comparison of the paper, new experiment to check the validity of the trained baselines**
>
> Thank you for your insightful review. We address each of your questions below. We provide an additional experiment serving as a sanity check for our choice of hyperparameters for the baselines. Our comments are divided into several posts because of the character limitations.
>
> ## 1. Experimental comparison
>
> ### Experimental setting
>
> In our paper, we have compared CAMEL with a number of baselines, including gradient-based meta-learning algorithms MAML and ANIL and the multi-environment context-informed network CoDA.
> We agree that our experimental conditions are not exactly the same as those of (Wang *et al.*, 2022) and (Kirchmeyer *et al.*, 2022). This is because **we do not address the same learning problem**. These papers focus on the problem of learning a function $f_\star$ representing a dynamical system by fitting the trajectories of the system. As a consequence, the training loss function is expressed in terms of the trajectory obtained by integration of the model, typically using Neural ODEs (Chen *et al.*, 2018). Although trajectory prediction is very rich and interesting, the trajectory integration step adds up a layer of complexity to the multi-task learning problem.
>
> We, on the other hand, chose to focus on the more fundamental problem where $f_\star$ is learned by regression. This is not only motivated by the fact that many physical systems are static (for example, the electrostatic field falls within this category, and could not be learned with a trajectory prediction approach), it is also the most general and natural framework for assessing the learning performances (including its expressivity and its computational cost) of a given architecture for learning a target function $f_\star$, regardless of the additional problem of integrating the dynamics into a trajectory.
>
> Therefore, we decided to run independent experiments on a number of systems representing various settings of regression, for which we compared our approach as fairly as possible to the other existing architecture, which we have reimplemented for static regression and carefully tuned. As we point out in the conclusion (Section 7), CAMEL can be extended to the setting of trajectory prediction for which identification of the physical parameters is also an important question. We are currently working on this extension and the results should be ready within weeks, in which case we can add them to an updated version of the paper.
>
>
> ### Choice of hyperparameters
>
> While we agree that comparing algorithms on different initial conditions may induce a bias related to the choice of hyperparameters, we would like to make it clear that we have conducted our experiments as fairly as possible. Hyperparameters such as the learning rates have been chosen on a test set in such a way as to give the best possible performance, for all architectures.
>
>
> **Additional experiment**
>
>
>  We present a new in-domain evaluation experiment showing that our choice of hyperparameters give similar performances for MAML and CAMEL in in-domain evaluation. The algorithms trained on multi-task data are evaluated **not on a new out-of-domain task, but on the training tasks**, albeit with different data points from the training set.
>
> Evaluation mean squared error
> |experiment|in-domain |out-of-domain (results presented in the submission)|
> |---|---|---|
> |MAML |8.1e-5 ± 2e-5|1.6e-1 ± 5e-2|
> |CAMEL|3.2e-5 ± 1e-5|1.0e-4 ± 5e-5|
>
>
> This experiment demonstrates that in a case more favorable to MAML's meta-learning, its performance is much closer to that of CAMEL and CoDA.
>
>
> We would also like to point out that the quantities reported on the right in Table 2 are not gradient step sizes, they are computation times, for training and during inference (a step size of order 30 or 10 would indeed be probably bad).

---

> > ### Author Response · Authors · 2023-11-15
> > **Answering to the questions about the missing references and about interpretability**
> >
> > ## 2. Meta-reinforcement learning
> >
> > We thank you for pointing out omitted related work and we will make sure to add it to the body of our paper in an updated version. We also highlight the differences between these approaches and ours.
> >
> > In (Nagabandi *et al.*, 2018), the forward dynamics is learned from multi-task data, as opposed to the inverse dynamics function learned in our approach. Although the learned model can then be directly used with a model-based control algorithm, no a priori structure of the learned function is known, and interpreting the task parameters and performing adaptive system identification therefore seems out of reach.
> >
> > In (Sodhani *et al.*, 2021), the control policy is directly meta-learned, in a model-free fashion. This method is arguably more flexible and the weights of the networks are interpreted statistically. However, it the learned parameters interpretable since the goal is not to learn a model of the system.
> >
> > ## 3. Interpretability
> >
> > We agree that "interpretability" is a big word, and that our approach provides only a partial answer to the question of model interpretability.  It is, however, a first step towards this goal, since our method allows the physical parameters of systems that vary from task to task to be identified, with applications to zero-shot adaptation and to online system identification in robotics for example.
> >
> > Identifying all the parameters of the learning model seems out of reach when using deep neural networks. An interesting and realistic research direction would be the use of invertible neural networks in $v(x;\theta)$, so as to identify not only the physical parameters but also the feature map shared across tasks.
> >
> > If necessary, we propose to change the title of our submission to "Towards Interpretable Meta-Learning of Physical Systems".
> >
> > ## References
> >
> > Wang, R., Walters, R., & Yu, R. (2022). Meta-learning dynamics forecasting using task inference. Advances in Neural Information Processing Systems, 35, 21640-21653.
> >
> > Kirchmeyer, M., Yin, Y., Donà, J., Baskiotis, N., Rakotomamonjy, A., & Gallinari, P. (2022, June). Generalizing to new physical systems via context-informed dynamics model. In International Conference on Machine Learning (pp. 11283-11301). PMLR.
> >
> > Ricky TQ Chen, Yulia Rubanova, Jesse Bettencourt, and David K Duvenaud. Neural ordinary
> > differential equations. Advances in neural information processing systems, 31, 2018.
> >
> > Nagabandi, A., Clavera, I., Liu, S., Fearing, R. S., Abbeel, P., Levine, S., & Finn, C. (2018, September). Learning to Adapt in Dynamic, Real-World Environments through Meta-Reinforcement Learning. In International Conference on Learning Representations.
> >
> > Sodhani, S., Zhang, A., & Pineau, J. (2021, July). Multi-task reinforcement learning with context-based representations. In International Conference on Machine Learning (pp. 9767-9779). PMLR.

---

> > > ### Comment · Reviewer_3DG3 · 2023-11-15
> > >
> > > Thank you for the response! Here are some of my quick initial thoughts to it:
> > >
> > > # Comparison with [Wang et al.](https://arxiv.org/pdf/2102.10271.pdf) (DyAd)
> > >
> > > >  This is because we do not address the same learning problem [...]  As a consequence, the training loss function is expressed in terms of the trajectory obtained by integration of the model,  typically using Neural ODEs (Chen et al., 2018). [...] We, on the other hand, chose to focus on the more fundamental problem where  is learned by regression.
> > >
> > > DyAd still seems more related than your response makes it seem. For example, as a first step it seems extremely easy to modify DyAd's model (and nothing else) to be linear in the context rather than the current non-linear architecture they have. If this is true, then evaluating this model comparison on exactly their experimental setup would be an extremely insightful ablation.
> > >
> > > Also, DyAd does /not/ integrate an ODE (their model also discretizes time) and they also learn by regression --- section 2.2 of their paper describes their full setup.
> > >
> > > I agree DyAd assumes knowledge of the contextual information and that another dimension of your approach is to learn the contextual information. This is an important difference. If the linear-context model works in their setting, it again could be very insightful to see how well you're able to recover the known contextual information from DyAd's setting if you assume it's not given.
> > >
> > > # Comparison with [Kirchmeyer et al.](https://arxiv.org/pdf/2202.01889.pdf) (CoDA)
> > >
> > > >  This is because we do not address the same learning problem [...]  As a consequence, the training loss function is expressed in terms of the trajectory obtained by integration of the model,  typically using Neural ODEs (Chen et al., 2018). [...] We, on the other hand, chose to focus on the more fundamental problem where  is learned by regression.
> > >
> > > I copy-pasted this again because I am not sure what you are referring to that integrates the ODE and does not do learning with regression. CoDA does /not/ integrate an ODE (their model also discretizes time) and they also learn by regression.
> > >
> > >
> > > # Hyper-parameters
> > >
> > > > We would also like to point out that the quantities reported on the right in Table 2 are not gradient step sizes, they are computation times
> > >
> > > What units are those times in? I assumed they were adaptation sizes because their values are just (1, 2, 10, and 30) with labels saying "training" and "validation"
> > >
> > > > While we agree that comparing algorithms on different initial conditions may induce a bias related to the choice of hyperparameters, we would like to make it clear that we have conducted our experiments as fairly as possible. Hyperparameters such as the learning rates have been chosen on a test set in such a way as to give the best possible performance, for all architectures.
> > >
> > > Did you also tune the gradient steps for training/validation? I noticed in another response you said:
> > >
> > > > We have omitted this experimental detail in our submission and will add it in the updated version. The number of inner gradient steps for training MAML and ANIL is equal to 1, in order to save computational time.
> > >
> > > It is widely-known that 1 gradient step for MAML/ANIL is not optimal. [This paper](https://arxiv.org/abs/1810.09502), for example, shows that a schedule over gradient steps during training is ideal. This still makes me believe the baselines are under-performing on these non-standard setups.

---

> > > > ### Comment · Reviewer_3DG3 · 2023-11-15
> > > > **On modeling forward dynamics vs inverse dynamics**
> > > >
> > > > Is there a reason why you decided to focus on inverse dynamics rather than the forward dynamics? It seems like the linear-context model could also be applied for the prediction of forward dynamics and compared to the other meta-learning approaches there.

---

> > > > > ### Author Response · Authors · 2023-11-16
> > > > > **Comparison with DyAD and CoDA**
> > > > >
> > > > > We thank you for your quick answer. We answer your questions in order, in two messages.
> > > > >
> > > > > ## Comparison with DyAd (Wang et al., 2022)
> > > > >
> > > > > We agree that Dyad's learning setting is more similar to ours, as the meta-model is trained by regression with discrete time steps. It would indeed be very interesting to see if its learning capabilities can be preserved while modifying its architecture to be linear in the context.
> > > > >
> > > > > We understand that this could be done by modifying the architecture so that the encoded vector representing the physical context is used only in the last layer of the predictor, so that the output is a linear function of the context. Did we correctly understand the modification that you suggest?
> > > > >
> > > > > Operating this modification and comparing it to the original model on the same setting would be a very interesting experiment, but it involves re-training the model, and potentially training the predictor and context encoder jointly, so that the learned contexts are meaningful in the embedding space.
> > > > > We are implementing it but generating the experimental data is currently an issue, as DyAd's code seems to be incompatible with the current version of PhiFlow. We are working on this issue and will try to produce the suggested experiments.
> > > > >
> > > > > ## Comparison with (Kirchmeyer *et al.*, 2022)
> > > > >
> > > > > What we mean by "the training loss function is expressed in terms of the trajectory obtained by integration of the model", is that CoDA is not trained to minimize a loss function of the form
> > > > > $$\sum_i \frac{1}{2} \Vert f(x_i ; \theta) - y_i \Vert^2,$$
> > > > > as in a classical regression task. For CoDA, the training loss is of the form
> > > > > $$\sum_i \frac{1}{2} \Vert \hat{x}(t_i) - x(t_i) \Vert^2,$$ where $\hat{x}$ is the integrated model, solving $\mathrm{d} \hat{x} / \mathrm{d}t = f(\hat{x}; \theta)$, and $x(t_i)$ are samples from the true trajectory.
> > > > >
> > > > > This is explained in details in section 4. [of their paper](https://arxiv.org/abs/2202.01889), in equation (10), and in the next paragraph:
> > > > > `To compute` $\hat{x}$ `, we apply for integration a
> > > > > numerical solver`.
> > > > > In the [official implementation of CoDA](https://github.com/yuan-yin/CoDA), the Forecaster module contains a numerical integration step using the torchdiffeq differentiable solver to produce the trajectories from the model (line 67 of file ode_model.py).
> > > > > Although we agree that time is discretized, the model appears only implicitly in the training loss through the integrated trajectory, not implicitly as in the classical regression loss with fixed points $x_i$ and targets $y_i$.
> > > > >
> > > > > Therefore, we argue that the prediction task in this paper is not the same as ours, since an additional layer of complexity is added to the model by the numerical integration. For the more fundamental problem of static regression, we have implemented CoDA with its hypernetwork architecture, and without the numerical integration layer, and we have compared it with the other baseline on an equal footing.
> > > > >
> > > > > CAMEL could be extended to the trajectory prediction setting, and it would indeed be interesting to compare both its prediction performances and its identification abilities on the same experimental conditions as CoDA. This is currently work in progress.
> > > > >
> > > > >
> > > > > ## Table 2
> > > > >
> > > > >
> > > > > The unit was chosen arbitrarily as the lowest measured time (in this case, that of CAMEL), for a clearer comparison between the baselines. The time for training CAMEL for 10000 gradient steps on a single CPU in the capacitor experiment was typically of the order of the minute. The time for adaptation is the time of solving least square regression, so it is in the fractions of second.
> > > > >
> > > > > To obtain the figures of Table 2, we have averaged the values over the experiments, and divided them by the lowest measured time in the baselines (the unit mentioned above). We will provide more details about this protocol in an updated version of the submission.

---

> > > > > > ### Author Response · Authors · 2023-11-16
> > > > > > **MAML hyperparameters and inverse dynamics**
> > > > > >
> > > > > > ## Hyper-parameters
> > > > > >
> > > > > > ### Tuning the gradient steps
> > > > > >
> > > > > > In our experiments, we have tuned the learning rates, both at training to stabilize the training dynamics and at inference time to make sure to correctly fine-tune the model on the adaptation set.
> > > > > >
> > > > > > At test time, we have also chosen an appropriate number of gradient steps to ensure that the adaptation loss is minimized.
> > > > > >
> > > > > > Setting to 1 the number of inner gradient steps for MAML might indeed be suboptimal for training. In our submission, this choice was imposed by computational constraints. Indeed, the number of gradients to be calculated is large in the capacitor example (where the grid is of high resolution), and the long training time makes it challenging to train MAML under the same conditions as the other approaches. Although increasing the number of inner gradient steps improves training stability, the computational cost is multiplied.
> > > > > >
> > > > > > We have conducted an additional experiment where the number of inner gradient steps for MAML is set to 5 instead of 1, in the electric charge system as before. The results are presented below.
> > > > > >
> > > > > > |experiment|in-domain |out-of-domain |time
> > > > > > |---|---|---|--|
> > > > > > |MAML, 5 inner steps |3.4e-5 ± 2e-5|1.6e-2 ± 7e-4|45|
> > > > > > |CAMEL|3.2e-5 ± 1e-5|1.0e-4 ± 5e-5|1|
> > > > > >
> > > > > >
> > > > > > Using additional inner steps indeed improves training as it reduces the test error of MAML by an order of magnitude and further closes the gap between CAMEL and MAML. However, the computational cost is also multiplied by 5. In the example of the capacitor, we go from a training time of a few minutes for CAMEL to a training time of tens of hours for MAML!
> > > > > >
> > > > > > There are arguably other operations than can be used to mitigate these costs and improve MAML's training. We believe that since all the architectures share the same underlying neural network, the performances of both approaches can be made even closer. However, this does not change the main conclusion of our work, which can be summarized as follows.
> > > > > >
> > > > > > Our approach enables efficient learning of physical systems, with a relatively simple architecture that leverage the structure of physical equations and requires less compute than gradient-based meta-learning approaches, both at training and at inference. These considerations are important for the applications that we have presented, such as robotics where the model must adapt in real-time (typically at 200 Hz), and gradient computations are often too costly. For these physical systems, we argue that the computational cost is as important as prediction performance. Furthermore, we have demonstrated that, unlike the baselines, CAMEL can efficiently interpret and identify the physical parameters, which is a valuable benefit that finds applications in online system identification for example.
> > > > > >
> > > > > > Strictly comparing the expressivity of the two architectures independently of the computational cost is an interesting problem which has been tackled from a theoretical point of view in (Wang et al., 2021), suggesting that they have similar learning capabilities. We have proposed an experimental approach confirming this theoretical insight on physical systems of interest, in the limit of a certain realistic computational budget. We agree that more extensive experiments on standard settings should be conducted in a future work to conclude on their expressivity from a pure learning viewpoint.
> > > > > >
> > > > > > ## Forward dynamics
> > > > > >
> > > > > > Forward dynamics could also be modeled and learned with CAMEL. The reason why we focus on inverse dynamics is because of the structure that it offers: the inverse dynamics function is linear in the inertial parameters. Leveraging this structure, learning the inverse function  ensures that these physical parameters can be identified, following Proposition 1 of our submission.
> > > > > > If we were to learn the forward dynamics instead, we would destroy this structure due to the presence of the inverse of the mass matrix. Therefore, learning the function would be harder and parameter identification would be out of reach. One of our primary goals is the interpretability of the learned models, including our original applications to online system identification in robotics (see Section 5.2 of our submission). These applications rely on the structure of the equations and could not be proposed if we learned the system's forward dynamics.
> > > > > >
> > > > > >  This point is explained in Example 4 of our submission and in Appendix B.2, and a more detailed derivation can be found in (Tedrake *et al.*, 2022), Chapter 18, section 18.2.2.
> > > > > >
> > > > > > ## References
> > > > > >
> > > > > > Wang, H., Zhao, H., & Li, B. Bridging multi-task learning and meta-learning: Towards efficient training and effective adaptation. In International conference on machine learning, 2021
> > > > > >
> > > > > > Russ Tedrake. Underactuated Robotics. 2022

---

> > > > > > > ### Comment · Reviewer_3DG3 · 2023-11-16
> > > > > > >
> > > > > > > Thank you for all of those clarifications! I am still positive on the paper and have updated my score from a 5 to a 6 (marginal accept) as these clarifications help me understand the difficulties of directly comparing to related work. The physical systems considered in this paper are nice new experimental settings for the meta-learning community to be thinking about. My hesitation in the paper is still due to there being no direct comparisons to existing closely-related meta-learning settings, such as the DyAd ones (which you understood my point correctly). This would be extremely convincing, and I would be willing to further increase my score with this.
> > > > > > >
> > > > > > > > On the MAML hyper-parameters/step sizes
> > > > > > >
> > > > > > > Thanks for the clarifications and more experimental details here too. Maybe I should have better-clarified earlier -- I'm also still in agreement with you all that even if MAML is well-tuned, the generalization performance should never be as great as a method like yours which is able to recover a more structured model such as yours that is closely related/identical to how the ground-truth system is structured.

---

### Official Review · Reviewer_rUF2 · 2023-11-01

**Soundness:** 3 good
**Presentation:** 3 good
**Contribution:** 3 good
**Rating:** 8
**Confidence:** 3

**Summary:**

This article presents a new meta-learning method, CAMEL, which uses affine task-specific parameters. It is motivated by the application of modelling physical systems, many of which are described in the article. CAMEL is demonstrated on these modelling examples and a robotic control task, with comparisons to state-of-the-art meta-learning methods MAML, ANIL, and CoDA. The interpretability of the method as a means of discover physical parameters is discussed.

**Strengths:**

The article is well-motivated and well-written. Both the motivation towards interpretable data-driven modelling of physical systems, and the motivation of simplifying meta-learning approaches like MAML, are well justified.

The proposed CAMEL method appears to be a simple but effective method for meta-learning. The overview of meta-learning approaches, comparing MAML, CoDA, and CAMEL, in section 2.2 provides a useful overview of the different methods. The use of different $w_t$ vectors for different environments is motivated in the text and suitable for physical applications, especially as $w_t$ can be learned used least squares.

The experimental evaluation of CAMEL is convincing, with a number of different physical modelling systems used for demonstration. CAMEL appears to match or outperform other meta-learning approaches on all tasks, although further details could improve the experimental evaluation (as detailed below). The code is provided and is simple to understand, using PyTorch for single-file implementations of CoDA, MAML/ANIL, and CAMEL.

**Weaknesses:**

The experimental results rely heavily on Table 2, however the experimental conditions for these results are not made clear. Specifically, the training details for MAML, ANIL, CoDA, and CAMEL are not fully detailed. The "Training" and "Adaptation" columns are not described, nor are their units present. The caption states that the table presents the "Average adaptation mean squared error and computational time," but the number of trials over which an average is taken, the standard deviation, the units of computational time, and how computational time was measured are not given. Statistical tests on the significance of values presented in Table 2 would be helpful.

The argument that CAMEL is an interpretable method relies on the idea that the weight vector $w_t$ learned on the training meta-dataset can correspond to physical parameters of a system. This is explained in Proposition 1, which assumes that the training loss is taken to 0. The authors do point out that identifying the meaning of each parameter, in other words, interpreting the parameters, is out of reach for black-box meta-learning architectures. However, the argument that CAMEL allows for such interpretation is not sufficiently demonstrated in my opinion. The example of learning an electric point charge system is given in section 5.1 and expanded in section B.5, however it relies heavily on the fact that CAMEL is able to learn the dynamics of the system, and not on the interpretability of the resultant parameters. Demonstrating how the learned physical parameters allow for an interpretation of the model and physical system, beyond the performance of CAMEL, would help make this argument. Physics informed neural networks (PINNs) share the same motivation and have been used to present interpretable results; their inclusion in this section and discussion would further improve the interpretability argument.

Lastly, the authors appear to only study the case in which $c = c_* = 0$. This is a large simplification. If Equation 3.1 is valid, I believe this means that $h(x;\theta_0) = 0$, which is quite a departure from other meta-learning methods. If this is the case, the authors should further justify it and potentially explore the case where $c \neq 0$. If it is not the case, then clarification is needed in section 4.3 as to when this simplification is made.

While the article is well-written, it suffers from two minor presentation weaknesses. The first is a reliance on popularity as justification; the popularity of a method does not justify its use. The sentence "Given the popularity and expressiveness of neural networks, incorporating multi-task learning into their gradient descent training algorithms is a major challenge" (2.2) is an example. The challenge of multi-task learning is not dependent on the popularity of neural networks, so why is it used as an opening clause? The second formatting issue is minor: works are cited in text when they should be cited in parentheses (although there are also proper uses of in-text citations). See, for example, this paragraph in section 4, where the Ljung citation is correct but the Nelles citation isn't:
"System identification and model identifiability are key issues when learning a system (Ljung, 1998). Although deep neural networks are becoming increasingly popular for modeling physical systems, their complex structure makes them impractical for parameter identification in general Nelles (2001)."

**Questions:**

How many trials were used to compute the averages in Table 2?

What do the "Training" and "Adaptation" columns correspond to in Table 2?

How many gradient steps were used in the various benchmarks?

Were the hyperparameters of the Adam optimizer the same for all benchmarks?

Could common physical parameters be learned in $v_*(x)$? What guarantees that $v_*(x)$ is task-agnostic?

Do the authors consider that their experimental results validate the hypothesis of equation 3.1?

---

> ### Author Response · Authors · 2023-11-15
> **Answering to the issues pointed out in the "weaknesses" section**
>
> Thank you for your time and thoughtful review. We address your comments below. Our answer is separated into several posts because of the character limitation.
>
> ## Table 2
>
> Our experimental results indeed lack some details. The number of experimental trials for evaluation is mentioned in the text body for Section 5.1 and in B.1.2 for the capacitor, but we have omitted to specify it for the other experiments. The standard deviations are reported in Table 3, in Appendix B.4. We shall update this section with the number of trials, and also report the latter in the legend Table 2. Regarding the computational time, we shall include additional details regarding the protocol that we used and the statistics of the measured time. Once measured and averaged, the unit was chosen arbitrarily as the lowest measured time (in this case, that of CAMEL) for a clearer comparison. The time for training CAMEL for 10000 gradient steps on a single CPU in the capacitor experiment was typically of the order of the minute.
>
> ## Interpretability
>
> In the case of the point charge system, we demonstrate the interpretability of the learned parameter by computing the identification error, *i.e.* by showing how close to the physical parameters (in this case the electric charges) the learned parameters are, which relies on CAMEL's ability to learn the system. We would like to point out that we further demonstrate the interpretability of the learned model on two major and novel applications.
>
> Firstly, the interpretation of the learned contexts allows for a zero-shot adaptation scenario (in the last paragraph of Section 4.), where the adapted model uses information about the physical parameters rather than measurements of the target function. This application is presented in both the charge experiment and in the capacitor experiment.
>
> Secondly, for robotic systems, the interpretability of the learned task parameters allows for the identification of the robot's inertial parameters, which is valuable in adaptive control. This is illustrated in Section 5.2.
>
> We will make sure to mention physics informed neural networks (Raissi *et al.*, 2019), for which interpretability is ensured by learning the solution of a PDE, while enforcing the physical constraints directly in the loss function.
>
>
> ## Proposition 1
>
> Regarding the case where $c, c_\star \neq 0$, it can be handled by augmenting  $\varphi$ and $v_{\star}$, and $w$ and $v_\star$ with an additional dimension, with the corresponding component of $v$ and $v_\star$ equal to $1$. The conclusion of Proposition 1 then applies wit the assumption of $\{v_\star(x^{(i)})\}$ spanning $\mathbb{R}^{n+1}$. Hence, the augmented physical parameters can be recovered up to a linear transform, meaning that $\varphi$ can be recovered with an affine transform. We have tackled the interpretability in the case $c, c_\star \neq 0$ experimentally in the capacitor system, where $c_\star$ is non zero a fortiori because the electrostatic field is linearized around a nonzero value, as is pointed out in the review.
>
> We will clarify this explanation in a revised version of the submission.
>
> ## Presentation issues
> We thank you for pointing out these two presentation weaknesses. We will correct them in an updated version of the submission.

---

> > ### Author Response · Authors · 2023-11-15
> > **Answering to the questions**
> >
> > ## Questions
> >
> > `How many trials were used to compute the averages in Table 2`
> >
> > As mentioned above, we shall gather these numbers for the various experiments in an appendix section. The number of test trials were 30 for the point charge, 50 for the cartpole and the robot arm, 15 for Upkie, and 5 for the capacitor.
> >
> > `What do the "Training" and "Adaptation" columns correspond to in Table 2?`
> >
> > "Training" refers to the average time required for training the architecture, relatively to the number of training gradient steps. "Adaptation" refers to the time required to fine-tune the trained model at inference for a fixed adaptation test, *i.e* to solve equation (2.5) of the paper. The measured time was averaged over the different systems.
> >
> > `How many gradient steps were used in the various benchmarks?`
> >
> > We have omitted this experimental detail in our submission and will add it in the updated version. The number of inner gradient steps for training MAML and ANIL is equal to 1, in order to save computational time. For training, the number of gradient steps ranges from 5000 to 30000 depending on the system and the architecture, and were chosen so as to observe convergence of the training loss. The number of adaptation gradient steps for MAML, ANIL and CoDA vary from 5 to 50 and are also chosen so as to observe convergence of the minimization equation (2.5) of the paper.
> >
> > `Were the hyperparameters of the Adam optimizer the same for all benchmarks?`
> >
> > We started with the same value for all the benchmarks, then adjusted if necessary to smooth out the convergence or to accelerate it, depending on the model and on the experiment. We will report these values in the updated version of the paper.
> >
> > `Could common physical parameters be learned in` $v_\star$ ? `What guarantees that` $v_\star$ `is task-agnostic?`
> >
> > Since $v_\star$ is assumed to be constant across tasks, the values of physical parameters in $v_\star$ do not vary across tasks. There is hence no hope for learning them without further assumption on the dependence of $v_\star$ with respect to these parameters. In the case of the pendulum, for example, the acceleration of gravity $g$ is a physical parameter of $v_\star$ that is constant across experiments because all the experiments are performed in the same reference frame, on earth. If $v_\star$ is assumed to be completely unknown, then guessing the value of $g$ from data is impossible. However, if we have prior information that $v_\star$ depends on the function $g \sin q$, then $g$ can be guessed by estimating $v_\star$ with Proposition 1 and plotting the learned function for example.
> >
> > The task-agnosticity of $v_\star$ is an assumption and amounts to assuming an affine parametrization of the target function across tasks. This is ensured globally in the robotics systems (as explained in Example 4) and in the point charge system, and locally for arbitrary systems that can be locally linearized, following (4.3).
> >
> > `Do the authors consider that their experimental results validate the hypothesis of equation 3.1?`
> >
> > The hypothesis (3.1) concerns the linearization of the model with respect to its parameters, when learning an arbitrary multi-task system. In the capacitor experiment with full displacement, no assumption is made whatsoever on the linearity of the target function. Yet, we observe that the learning performance of CAMEL matches the performance of the other baselines. This suggests that the task dependence of the multi-task predictor can somehow be expressed as a linear function of some task weights, and hence suggests that (3.1) is valid.
> >
> > The answer above is qualitative, and could be supplemented by a more quantitative approach. For example, we could experimentally calculate the Jacobian (with respect to $\theta$) of the trained models MAML, ANIL and CoDA using automatic differentiation, and thus their linear approximation (3.1). By calculating the relative deviation of this tangent at $\theta_0$ (which is computed explicitly in all these architectures) from the function and by averaging the error over $x$ and $\theta$, we should be able to conclude quantitatively as to the validity of the approximation on our different physical systems.
> >
> >
> > ## References
> >
> > Raissi, M., Perdikaris, P., & Karniadakis, G. E. (2019). Physics-informed neural networks: A deep learning framework for solving forward and inverse problems involving nonlinear partial differential equations. Journal of Computational physics, 378, 686-707.

---

### Author Response · Authors · 2023-11-22
**We uploaded a revised version taking into account your comments**

Dear reviewers, we thank you for taking the time to review our paper and for your insightful remarks. We have uploaded a revised version of the manuscript taking your comments into account, with the major changes and additions higlighted in blue and deletions highlighted in red. The major changes include a better explanation regarding the novelty of our approach and its comparison to previous work, and the addition to the paper body of the experiments that we described in our responses to reviewers 3DG3 and X11Z.

### Change in notation
In addition to reformulations and minor changes, we have made the following notation change, based on the comments of reviewer X11Z, to improve clarity:
$$ v_\star \longrightarrow \nu$$
starting in equation (4.1), page 5.

## Page limit

We are aware that the current version exceeds the 9-page limit. This is due to the presence of highlighted deletions, for greater clarity. By removing these, we can bring the main text back within the 9-page limit.


Please let us know if you have further remarks or questions.

---

### Meta-Review · Area_Chair_MzED · 2023-12-08

**Metareview:**

This paper studies meta-learning for physical systems. It learns a feature representation that is common across tasks and a linear embedding/task weight for each task. This leads to strong interpretability and generalizability, as is demonstrated on several problems from physics. There was a lot of discussion for this paper. Reviewer X11Z criticized the lack of methodological novelty but agreed on the reframing the authors proposed in response, increasing their score to the borderline region (with other reviewer in favor of acceptance). Overall, this is a very interesting paper and I follow the consensus towards accepting it.

**Justification For Why Not Higher Score:**

The methodology is only somewhat novel.

**Justification For Why Not Lower Score:**

It is a simple and effective meta-learning approach, with strong results in problems from physics.

---

### Decision · Program_Chairs · 2024-01-16

Accept (poster)